# Influence of Heating–Cooling Regime on the Engineering Properties of Structural Concrete Subjected to Elevated Temperature

Daniel Paul Thanaraj [1], Tattukolla Kiran [2], Balamurali Kanagaraj [2], Anand Nammalvar [2,*], A. Diana Andrushia [3], Beulah Gnana Ananthi Gurupatham [4] and Krishanu Roy [5,*]

1 Department of Civil Engineering, KMEA Engineering College, Kochi 682030, India
2 Department of Civil Engineering, Karunya Institute of Technology and Sciences, Coimbatore 641114, India
3 Department of ECE, Karunya Institute of Technology and Sciences, Coimbatore 641114, India
4 Division of Structural Engineering, College of Engineering Guindy Campus, Anna University, Chennai 600025, India
5 School of Engineering, The University of Waikato, Hamilton 3216, New Zealand
* Correspondence: nanand@karunya.edu (A.N.); krishanu.roy@waikato.ac.nz (K.R.)

**Abstract:** Structural concrete has become a highly preferable building material in the construction industry due to its versatile characteristics, such as workability, strength, and durability. When concrete structures are exposed to fire, the mechanical properties of concrete degrade significantly. The research on the residual mechanical properties of concrete after exposure is necessary, particularly for the repair and rehabilitation of concrete elements and for the stability of the infrastructure. Factors, such as the grade of concrete, the effect of temperature exposure, and rapid water cooling, affect the residual strength characteristics of concrete. Considering these factors, the present investigation evaluates the mechanical properties of concrete using different grades, such as those ranging from 20 to 50 MPa, with an increment of 10 MPa. The specimens were exposed to different durations of fire from 15 to 240 min, following the standard rate of heating. A loss of strength was observed after fire exposure for all the grades of concrete. The rate of reduction in tensile and flexural strengths of the concrete was greater than that of compressive strength. The experimental results also showed that the strength reduction is greater for M50 than M20 concrete concerning the duration of heating. A microstructure evaluation confirmed the extent of damage to concrete under varied temperature conditions.

**Keywords:** concrete; elevated temperature; water cooling; stress–strain; residual strength

## 1. Introduction

Concrete is a composite mixture that comprises a binder, filler, and aqueous solutions. In the world at present, concrete plays a significant role in developing civil infrastructures due to its superior mechanical and durability performances [1]. Fire has always been an incessant threat to the stability of a structure. It represents a vulnerability factor and a great hazard that can wreak havoc on buildings and civil infrastructures. Countless fire accidents periodically occur around the world and lead to the degradation of the essential qualities of a sound infrastructure. The load-carrying capacity of a building decreases significantly and leads to the collapse of the structure due to the degradation of the strength properties of the building materials [2]. When the concrete is subjected to temperature exposure, its hardening performance tends to drastically degrade. Therefore, there is a deterioration in the quality of the concrete [3,4].

The compressive strength of concrete is one of the primary factors used in the design of reinforced concrete structures. At the given temperature conditions, the compressive strength of concrete is based on the w/c ratio, cementitious material, aggregate type,

and curing conditions [5]. At a higher temperature, a greater reduction in the tensile strength and flexural strength of concrete can be observed, compared to its compressive strength [6]. The decline in the strength of the concrete is due to the decomposition of CSH gel, degradation of calcium hydroxide ($Ca(OH)_2$), decomposition of cement paste, and deformation of aggregates [7]. In recent years, there has been a considerable increase in structural fire engineering areas, which commonly deal with mix types, heating time, testing methods, adding fibers to the mix, and maximum temperature level [8].

During fire exposure, the concrete compressive strength (CS) does not become altered up to a temperature of 300 °C. When the temperature exceeds 300 °C, drastic changes occur in the critical mechanical properties. The strength degradation of concrete mainly occurs due to thermal incompatibility caused by temperature variations due to the presence of pore water [6]. The deterioration of the aggregates, decomposition of the hydrated pastes, and the thermal incompatibilities between the precursor and aggregates lead to strength disintegration [9]. The concrete's physical structure and chemical composition noticeably change with its exposure to high temperatures. The water molecule chemically bound to CSH gel phases becomes dehydrated and is more noticeable at temperatures higher than 110 °C [10]. Due to the degradation of hydrated gel phases and thermal expansion, internal stresses are developed. When the temperature exceeds 300 °C and above, micro-cracks are induced on the surface of the specimen. On increasing the temperature beyond 500 °C, calcium hydroxide is dissociated and results in concrete shrinkage [11].

Since concrete is a heterogeneous mixture, the constituents present inside the matrix influence its fire-resistant performance. Some of the other factors that influence the fire-resistant performance are binder content, w/c ratio, heating duration, and cooling type. Studies conducted on the fire-resistance properties of concrete reveal that changes in its mechanical properties play a significant role during temperature exposure. In recent studies, many fire-retardant materials were employed to safeguard the structures [12]. Fire-resistance design and construction are the primary requirements in atomic/nuclear and power plant structures [13].

Normal strength concrete (NSC) was primarily used in many previous studies as a research area [5,14–16]. Studies related to behavioral changes in different grades of concrete mixes under elevated temperature exposure are essential to understand their levels of degradation to enhance their performance during fire accidents. Studies conducted on concrete structures under elevated temperatures are essential in ascertaining the stability of a structure, because they help structural engineers to make the correct decision. The analysis report assists engineers in deciding whether an infrastructure needs to be retained or demolished. Recently, high-strength concrete (HSC), fiber-reinforced concrete (FRC), and lightweight aggregate concrete (LWC) have been used in high-temperature tests. Explosive spalling was the main problem in HSC [17]. It is reported in the literature that explosive spalling occurred at 300 °C for lower levels and the higher levels at 650 °C. LWC produces better results than HSC and NSC [18]. When the concrete is exposed to elevated temperatures, its internal microstructure is distorted and results in a drastic reduction in the strength and durability of the concrete. The structural and thermal behaviors of the varied grades of concrete subjected to different temperature loads will present an overview of how they respond to elevated temperatures. Additionally, it becomes a prerequisite to study and understand the behavior of structural materials subjected to fire and their mechanical response to heating.

After conducting a detailed literature study, it was observed that studies related to the residual and loss of mechanical characteristics of traditional concrete, HSC, and FRC have been extensively performed. The present experimental investigation thoroughly examines the influence of temperature exposure on the hardening performance of concrete specimens cast with various strength grades. Additionally, almost all the studies considered a natural air-cooling method to cool the exposed concrete specimens. The extent of damage in the fire-affected concrete depended on the magnitude of temperature, concrete mix, and cooling type adopted for extinguishing the fire. During fire accidents, generally, fire

on concrete structures is quenched by water extinguishers. Thus, estimating the damage level of water-cooled concrete structures is vital to evaluate the residual strength for repair and rehabilitation works. However, the experiments conducted on the fast-cooling (water-cooling) effect on physical, mechanical, and microstructure properties are minimal. The present study aims to create a database for the residual hardening performance of concrete subjected to standard fire conditions and to anticipate the expected changes in the mechanical behavior of concrete exposed to very high temperatures followed by fast cooling procedures.

In the research phase, various grades of concrete, such as M-20, M-30, M-40, and M-50, were developed. There is a scarcity of fundamental knowledge on the influence of standard fires on the thermal characteristics of concrete, notably for concrete with varying strength grades. While the knowledge and experience concerning various types of concrete behavior at ambient and increased temperatures are well established in the literature, the effect of water cooling on concrete with varied grades subjected to fire requires further investigation. The residual hardening performance, such as stress–strain behavior, tensile strength (TS), flexural strength (FS), and compressive strength (CS) of concrete, were also assessed. A numerical relationship was developed between CS to the TS and FS of the concrete. The results for the residual strength database with a related key influencing parameter are helpful to construction engineers. Since the residual hardening performance of water-cooled specimens significantly differs from air-cooled specimens, this data will help to formulate structural fire standards for better resistance properties. The data on the physical, mechanical, and microstructure properties of water-cooled specimens may be relevant in predicting the evacuation time of humans before structural collapse occurs during building fires.

## 2. Experimental Program

### 2.1. Materials

Ordinary Portland Cement (OPC) grade 53 was used in the study to prepare all concrete specimens. River sand with a particle size smaller than 4.75 mm was employed as a fine aggregate, following IS 383 (2016) [19]. The fine aggregates belonging to zone II were identified for casting. Crushed granite was employed as the coarse aggregate (CA) in which stones of varying sizes between 20 to 12 mm were downgraded. Figure 1 presents the particle size of filler materials. The study used portable water for mixing, casting, and curing the concrete specimens. Table 1 illustrates the material characteristics used in the concrete mixture.

**Table 1.** Physical properties of materials.

| Material | Density (kg/m$^3$) | Specific Gravity | Water Absorption |
|---|---|---|---|
| Cement | 1437 | 3.16 | - |
| Fine aggregate | 1651 | 2.71 | 0.42 |
| CA | 1798 | 2.87 | 0.56 |

### 2.2. Mix Proportion and Specimen Fabrication

Table 2 highlights the mix recipe that was adopted in this study. The concrete mix proportions of 20 to 50 MPa were designed as per 10262 (2019) [20]. Various trials were conducted to achieve the concrete's targeted fresh and hardened properties. The experimental study utilized a polycarboxylate ether-based super plasticizer (SP) to attain fresh properties of M-40- and M-50-grade concrete specimens. The target slump value was fixed at 125–150 mm for all the grades.

Specimen dimensions of 500 × 100 × 100 mm were tested to achieve the FS of concrete. For the CS of concrete, cube specimens 150 mm in size were used and tested as per ASTM C39 [21]. For the TS of concrete, cylindrical specimens with a diameter size of 150 mm and 300 mm in height were employed and tested as per the code guidelines [22]. Specimens were cast using a mixer drum and the concrete mixtures were placed in their respective

molds. The freshly prepared mix was allowed to set at room temperature for 24 h. Then, the specimens were transferred to the curing tank to initiate the hydration reaction process. The specimens were then removed from the curing tank and the mechanical properties of the concrete were checked, as presented in Figure 1a. Table 2 presents the mix details of the concrete.

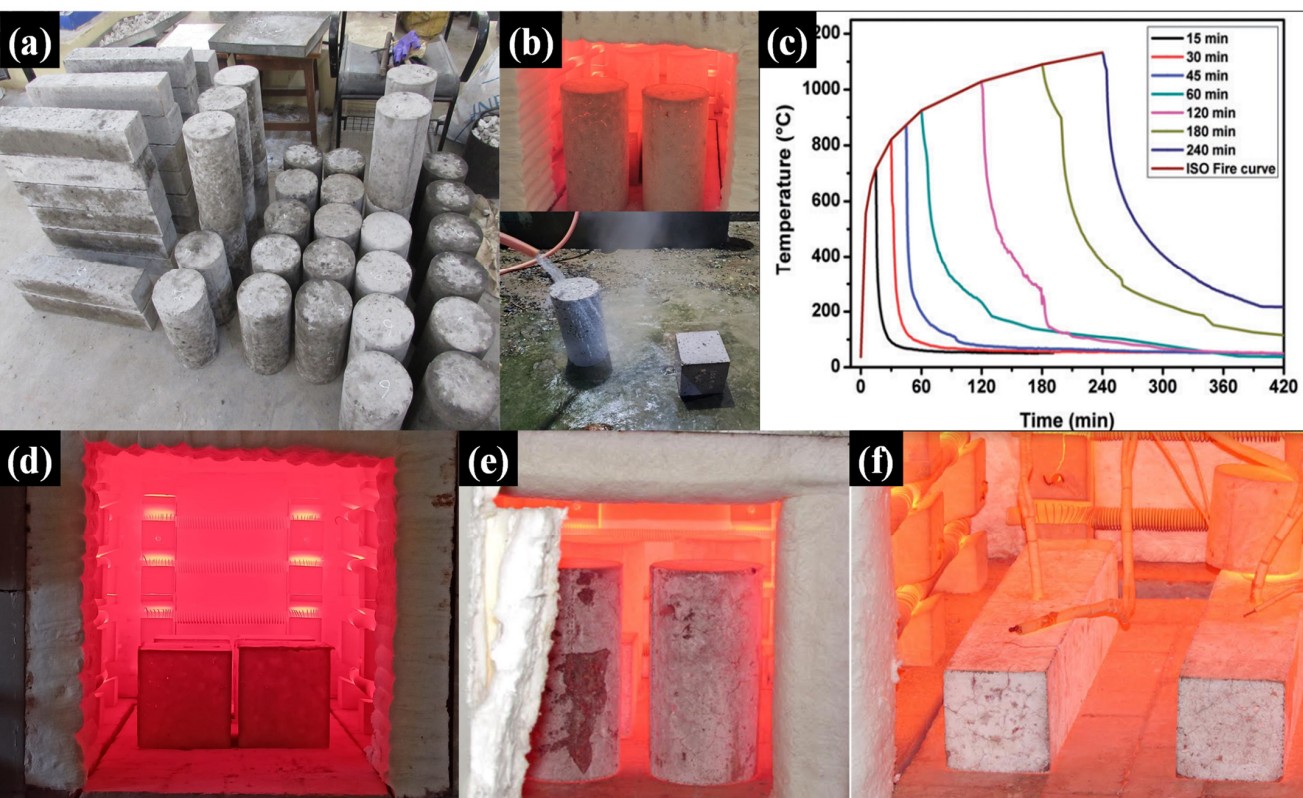

**Figure 1.** Images of (**a**) test specimens, (**b**) heating and water cooling, (**c**) heating–cooling cycle of concrete specimens, (**d**) cubes under elevated temperature, (**e**) cylinders under elevated temperature, (**f**) beams under elevated temperature.

**Table 2.** Details of design concrete-mix proportions.

| Material | M20 | M30 | M40 | M50 |
|---|---|---|---|---|
| OPC (kg/m$^3$) | 321 | 381 | 412 | 465 |
| River sand (kg/m$^3$) | 850 | 803 | 788 | 773 |
| Coarse aggregate (kg/m$^3$) | 1034 | 1088 | 1158 | 1245 |
| Water-to-cement ratio | 0.58 | 0.50 | 0.38 | 0.34 |
| Chemical admixture (l/m$^3$) | - | - | 4.94 | 5.59 |

*2.3. Elevated Temperature Test*

An electrical furnace was used to study the fire performance of the concrete specimens. The overall size of the furnace was $750 \times 500 \times 500$ mm. All concrete specimens were kept in the furnace and heated as per the guidelines provided in ISO 834 [23]. Following the heating test, all the specimens were cooled with a water spray. K-type thermocouples were used to monitor the surface and core-temperature penetration values of the concrete. The water-cooling process of the concrete specimens is illustrated in Figure 1b. The heating–cooling curve is presented in Figure 1c. Figure 1d–f shows the cube, cylinder, and beam specimens under elevated temperatures.

### 2.4. Hardened Properties of Concrete

After the cooling treatment, experiments were conducted to obtain the concrete's stress–strain behavior, FS, CS, and TS. An average of three samples was considered in the present investigation to achieve accuracy. A detailed experimental procedure is elaborated below. The test images are illustrated in Figure 2a–c.

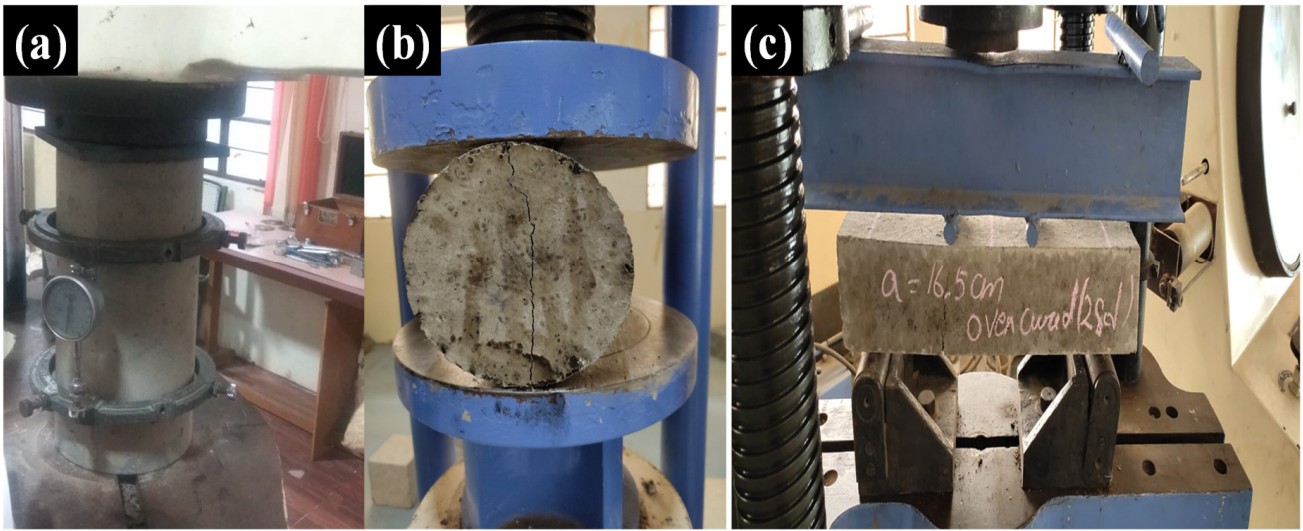

**Figure 2.** Images of (**a**) CS, (**b**) TS, and (**c**) FS tests of concrete.

#### 2.4.1. Ultrasonic Pulse Velocity Test

A UPV test was performed to analyze the homogeneity and pore density of the concrete. Concrete cube specimens were preferred to conduct the UPV. The direct UPV test method was chosen for better results. The experiment was conducted as per IS 516 (2019) [24]. A unipan 543 digital instrument and 40 kHz point ultrasonic heads were used for the test. The plus velocity was estimated based on the specimen's dimension and measured transit time.

#### 2.4.2. Stress–Strain Behavior of the Concrete Test

The stress–strain behavior of the concrete was evaluated using a cylinder-specimen dimension of 150 mm in diameter and 300 mm in length. The experiment was performed by using the universal testing machine. The equipment is well-designed to record the failure load of concrete and the stress–strain behavior of test specimens. The testing program was performed according to IS 516:2004 [24]. Figure 2a shows the testing of cylinder samples.

#### 2.4.3. Tensile Strength Test

The experiment was conducted to evaluate the tensile strength of concrete as per the guidelines of IS 5816:2004 [25]. The cylinder concrete specimen having a diameter of 150 mm and length of 300 mm was used for testing. The tests were performed using CTM equipment and the load was applied at a rate of 2 N/mm$^2$/min. The test readings were noted until specimen failure. Figure 2b shows the testing of samples for obtaining the desired tensile strength values.

#### 2.4.4. Flexural Strength Test

A detailed experiment was conducted to evaluate the flexural strength of the concrete. Prism specimens of 100 mm in width, 100 mm in depth, and 500 mm in length were used for the testing. The test standards were followed as per IS 516:2004 [24]. Samples were tested under the two-point loading setup. A constant load was applied at a constant rate of 180 kg/min, until the failure of the specimen. Figure 2c shows the flexural strength test of the concrete prism.

## 3. Results and Discussion

### 3.1. Ultrasonic Pulse Velocity (UPV)

Figure 3 shows the average UPV values of concrete specimens before and after temperature exposure for M-20 to M-50 concretes. The results were compared as per the IS 13311 [26] guidelines. The test results show that the UPV values increase with an increase in the grade of M-20 to M-50 concretes due to the concrete's higher-density structure. In the case of temperature-exposed specimens (15 and 30 min), they presented an excellent performance. At 45 min, M-20 and M30 specimens exhibited doubtful quality. However, the M-40 and M50 specimens were good at the same temperature exposure. After 60 and 120 min of exposure, the test result shows a poor performance. The specimens exposed for 180 and 240 min presented no results. This was due to the several surface cracks, voids, and damage present in the M-20- to M-50-grade concrete. Morphological changes, such as wider thermal cracks and poor CSH gel quality, led to a decrease in UPV values, and similar observations were also noted by Kanagaraj et al. [27].

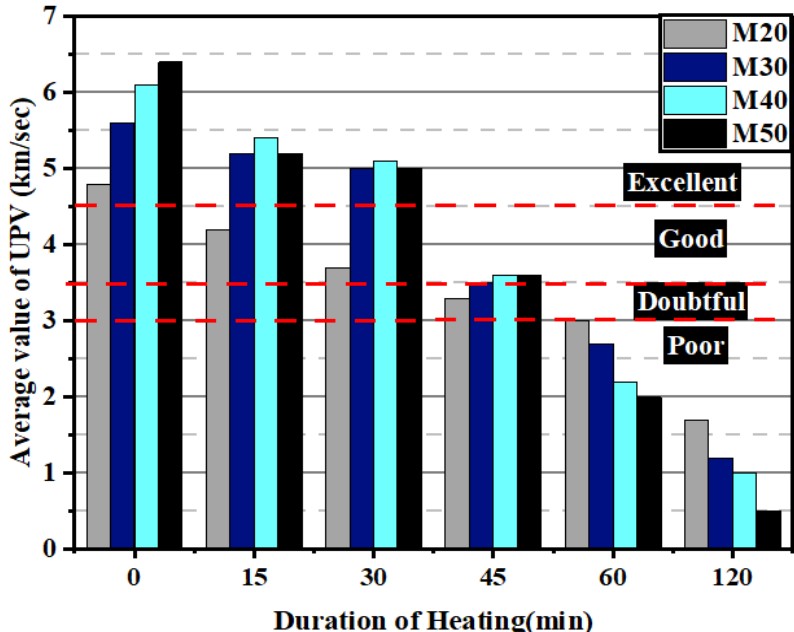

**Figure 3.** UPV values of M-20 and M-50 concrete specimens following elevated temperature exposure.

### 3.2. Stress–Strain Behaviour

Under uniaxial loading, the stress–strain behavior presents an idea of the concrete's mechanical response, including its stiffness [28]. As the concrete is exposed to heating, it becomes brittle and loses its stiffness (or modulus). The stress–strain curve of the reference and heated concrete for strength grades M-20–M-50 is shown in Figure 4. A computerized digital universal testing machine was employed to record the stress–strain points at every loading stage until failure. As the heating intensity increased, the CS decreased while the compressive strain increased. This behavior was different for the fire-affected concrete with a different grade. A notable difference in the stress capacity and strain level was observed between the M-20 and M-50 concrete specimens.

The ascending curves of the heated and unheated concrete specimens are different. The stress–strain plots for all the strength grades exposed to heating and cooled down by spraying water are shown in Figure 4. An increase in the intensity of heating decreased the peak values of the stress and strain. Contrastingly, the stiffness of the concrete started to decline with the increase in the heating temperature. Researchers [29] observed that at a heating temperature <300 °C, there was no sign of alteration in the stress–strain curve compared to the unheated specimens. The degradation of the concrete samples was observed to gradually increase upon increasing the heating duration to 45 min, and the

stress–strain curve became flat. The elastic modulus and CS simultaneously decreased, and the ultimate and peak strains rapidly increased. For durations of 120, 180, and 240 min, the descending curves of all grades of concrete were observed to flatten. There was an increasing trend for strain corresponding to peak stress. The strains corresponding to peak stress for 60 and 120 min of heating were observed to be ten and fifteen times that of the strain at room temperature, respectively. The specimens of all grades heated for 180 and 240 min showed higher strain values. Up to 400 °C, there was no significant change in the strain corresponding to the peak strength for all four types of HSC.

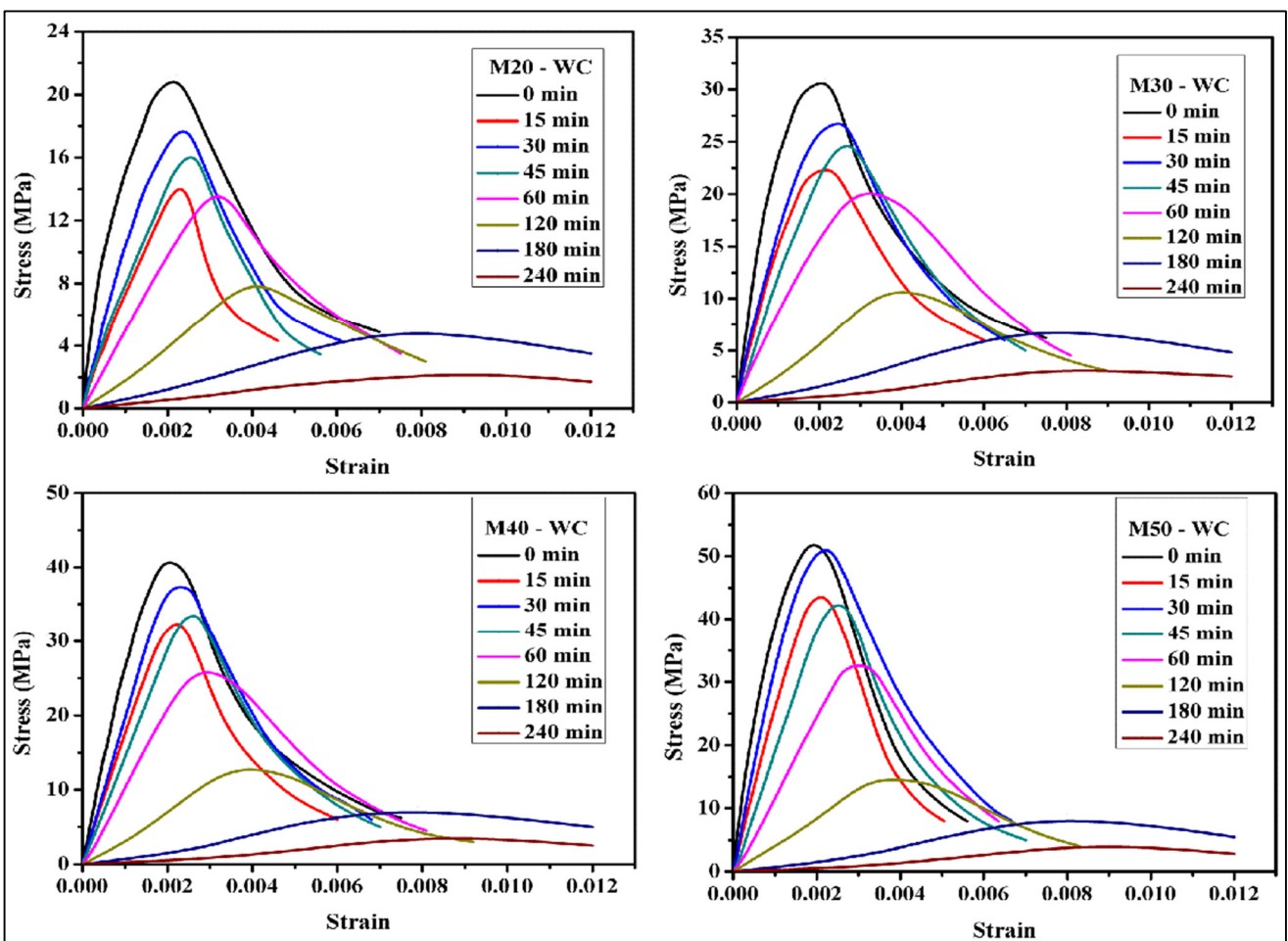

**Figure 4.** Stress–strain behavior of water-cooled specimens.

### 3.3. Residual Compressive Strength

In lower-grade concrete with a higher water content, the water evaporates and develops more pores when heated. This naturally weakens its internal microstructure. Between 15 and 30 min of heating, all the specimens showed a slight increase in strength. M-50 concrete specimens were observed to have the maximum strength gain. M-40 and M-50 grades slightly enhanced their strength by 1.5% and 5.4% of the original strength. Therefore, within 30 min of exposure, M-50 grade was observed to exhibit a strong performance. This can be attributed to the fact that the water-to-cement ratio of M-50 is low as it is less hydrated. Consequently, a greater quantity of unreacted cement particles is available for rehydration under 30 min. Over 30 min, a greater reduction in stress occurred in M-50 concrete specimens, compared to other specimens [30]. This decrease in strength was due to the dehydration and decomposition of the internal chemical structure in the cement paste as the exposure duration and temperature increased [31,32].

Beyond 45 min of exposure, M-20 and M-50 exhibited a loss of CS by 9.5% and 15.2%, respectively. At 60 min of exposure, the loss of strength was observed to be 23.4% and 29.6% for M-20 and M-50 concrete specimens, respectively as shown in Figure 5. At this stage, strength loss was due to pore coarsening. The vaporization of water inside the concrete was the prime reason for the reduction in strength. High-strength concrete has fewer pores; hence, the dissipation of gases during heating becomes more difficult. During heating, the vapor pressure exceeds the ultimate TS and thus results in the widening of the cracks. In the case of low-strength concrete subjected to high-temperature exposure, the pores inside the concrete become interconnected, which results in the formation of capillary tubes. This leads to the dissipation of water vapor. The size of the pores increased at rates exceeding 120 min of heating. A loss of strength was observed between 88 and 93.2% at 240 min of heating. At 240 min, the strength reduction rate was almost the same for all grades of concrete. From 180 to 240 min of exposure, the effect of heating duration and temperature surpassed the effect of CS of concrete on the strength loss of specimens. Generally, the collapse and dehydration of CSH gel occurred beyond 500 °C. Additionally, remarkable chemical changes also occurred in the cement paste. Usually, over 900 °C, a complete decomposition of concrete occurs and then the concrete loses its strength, as described by Topcuet et. al [33]. The observed standard deviation for M20 and M50 concrete specimens was 0.59 and 0.84, respectively.

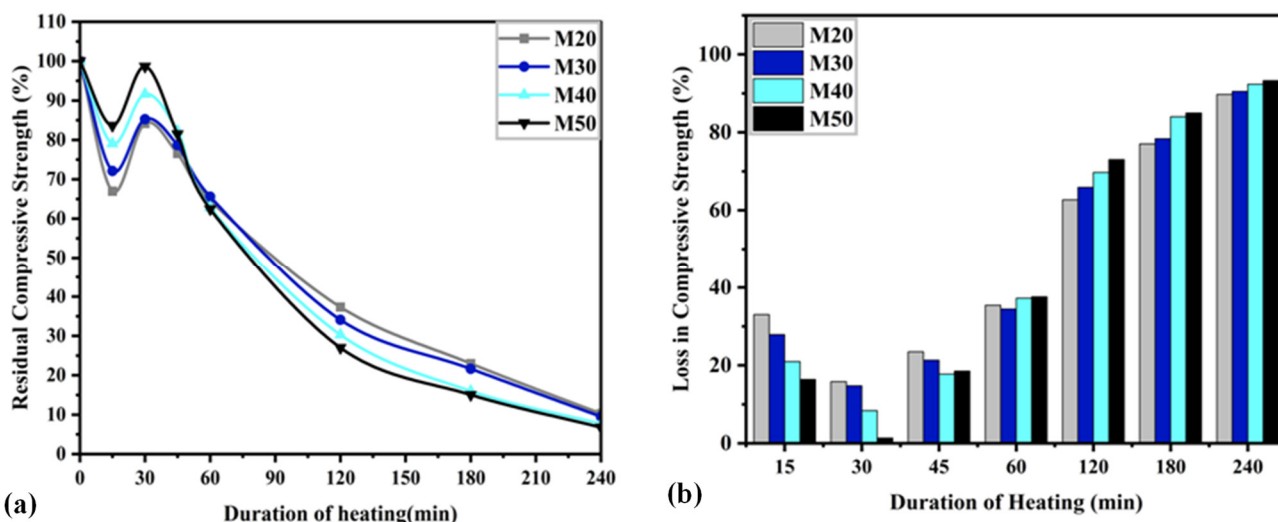

**Figure 5.** (**a**) Residual CS of concrete, (**b**) loss of CS of concrete.

*3.4. Correlation between Compressive Strength and UPV of Concrete*

Figure 6 represents the polynomial relationship between the compressive strength and UPV of concrete based on the fitting curve subjected to elevated temperatures. As the compressive strength of concrete increased, the UPV of concrete also increased. As the duration of heating and temperature increased, the compressive strength and UPV values drastically decreased. The proposed equations were observed to be excellent with high regression values.

*3.5. Tensile Strength*

The higher TS was attributed to the crack formation in the concrete. The higher TS enables a resistance of crack propagation developing in the concrete due to internal water pressure when exposed to elevated temperatures [34]. Figure 7 a, b depicts the residual and loss of TS of concrete exposed to different temperature regimes. An increase in temperature exposure significantly decreased the residual TS for all the strength grades. The residual TS values of different strength grades exposed to 15 min (718 °C) durations of heating and cooling by spraying water were 80.2%, 76.9%, 71.2%, and 68.5% of the initial strength for M-20- to M-50-grade concrete, respectively. The residual TS values of different strength

grades exposed to 30 min (821 °C) durations of heating and cooling by spraying water were 69.3%, 64%, 59%, and 54.1% of the initial strength for M-20 to M-50 samples. At 45 min (873 °C) duration of heating and cooling by spraying water, M-20 to M-50 samples had residual strength values of 58.4%, 52.3%, 43%, and 24.1% of the initial strength, respectively. The retained strength values of various strength grades beyond 60 min (925 °C) durations of heating and cooling by spraying water were 41.1%, 33.8%, 28.2%, and 24.1% of the initial strength for M-20 to M-50 samples. At 90 min of (977 °C) fire-exposure time, the retained/residual strength values of various strength grades of concrete specimens cooled by spraying water were 30.2%, 26.1%, 23.4%, and 20.3% of the initial strength for M-20 to M-50 samples, respectively. Furthermore, the strength slowly decreased after 90 min of heating (977 °C), and the residual tensile strengths reduced to 6.9%, 5.3%, 4.3%, and 3.7% of the initial strength for M-20 to M-50 samples, respectively, at 240 min of exposure. Lower grades of concrete showed higher resistance to heat and thus a smaller reduction in TS compared to higher grades of concrete that exhibited an additional loss of strength. This may be due to thermal shock (stresses) and steep thermal gradients developing in the concrete during heat exposure. It has been speculated [35–39] that the decreased porosity of concrete with higher grades blocks the release of vapor. Therefore, the vapor-pressure stress in the interior region of the concrete surpasses the TS of concrete, which ends up producing a more micro- and macrocracks [30,40,41]. The observed standard deviations for M20 and M50 concrete samples were 0.11 and 0.07, respectively.

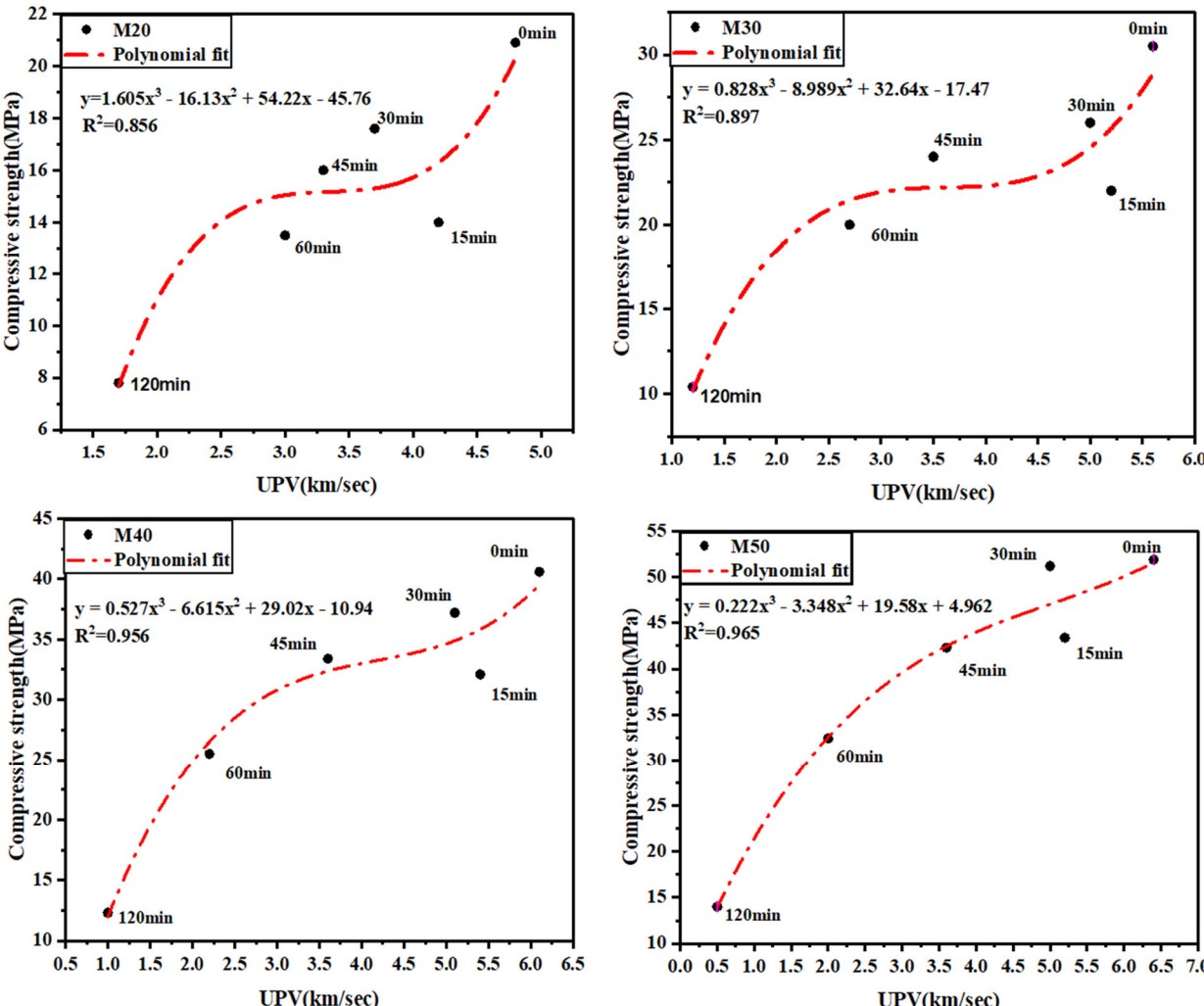

**Figure 6.** Correlation between compressive strength and UPV of M-20, M-30, M-40, and M-50 concrete samples.

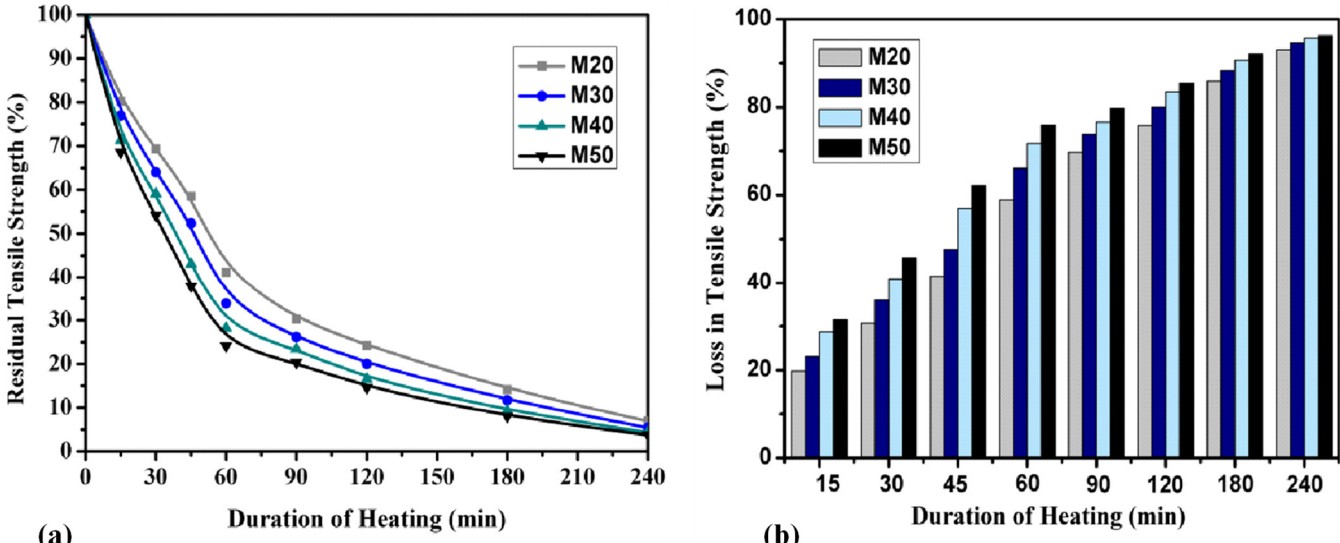

**Figure 7.** (**a**) Residual TS of concrete specimens, (**b**) loss of TS of concrete specimens.

*3.6. Flexural Strength*

The FS of specimens is directly proportional to the CS of concrete. A similar observation was made in the case of the residual FS of concrete. However, the reduction rate of FS was stronger than the residual CS of concrete. Figure 8a,b show the residual and loss of FS of concrete exposed to standard fire. Trials have been conducted to identify the residual FS of concrete specimens exposed to different temperatures for various heating and water cooling durations. At the beginning (up to 15 min), the reduction in FS was minimum. Later, the reduction in FS increased along with the increase in the heating duration. The retained FS of concrete with various strength grades cooled by water after exposure for 15 min was observed to be between 82% and 55% of the original strength (M-20 to M-50). Although all grades of concrete specimens gained CS at 30 min of fire exposure, a greater loss was observed for FS at this exposure duration. Losses in FS are attributed to cracks that degrade the cross-sectional area, and tensile stress causes cracks to widen [42].

Residual FS decreased between 15 and 60 min of exposure to the various concrete grades. The residual FS rates of different concrete grades cooled by water after being exposed to 30 min (821 °C) of heating were 60%, 50.6%, 42.7%, and 30.3% of the initial strength for M-20 to M-50 grades of concrete, respectively. At 60 min (925 °C) of heating, the residual FS rates of different concrete grades cooled by water were 32.2%, 22.6%, 16.0%, and 10.9% of the initial strength for M-20 to M-50 grades of concrete, respectively. The strength reduced further upon gradually being heated during 60–120 min (925–1029 °C). The residual FS rates of different concrete grades cooled by water after being exposed to 120 min (1090 °C) of heating were 15.6%, 10%, 7.2%, and 1.9% of the initial strength for M-20 to M-50 grades of concrete, respectively. The retained FS of the specimens exposed to 180 and 240 min was observed to be zero, as these specimens were damaged during the heating process. The retained FS was lower than the retained CS for all the strength grades in all the heating ranges. Heated specimens under flexural loading were more sensitive to cracking than compressive loading. Therefore, crack formation occurring under flexure was observed to be more critical than compression [43,44]. An increase in temperature exposure increased crack propagation; this might be attributed to the degradation of CSH gel phases [45]. The reduction in FS was greater for specimens with a higher grade than specimens with lower grades at all heating durations, and an additional strength decrease was also noticed. The observed standard deviations for M20 and M50 concrete samples were 0.26 and 0.13, respectively.

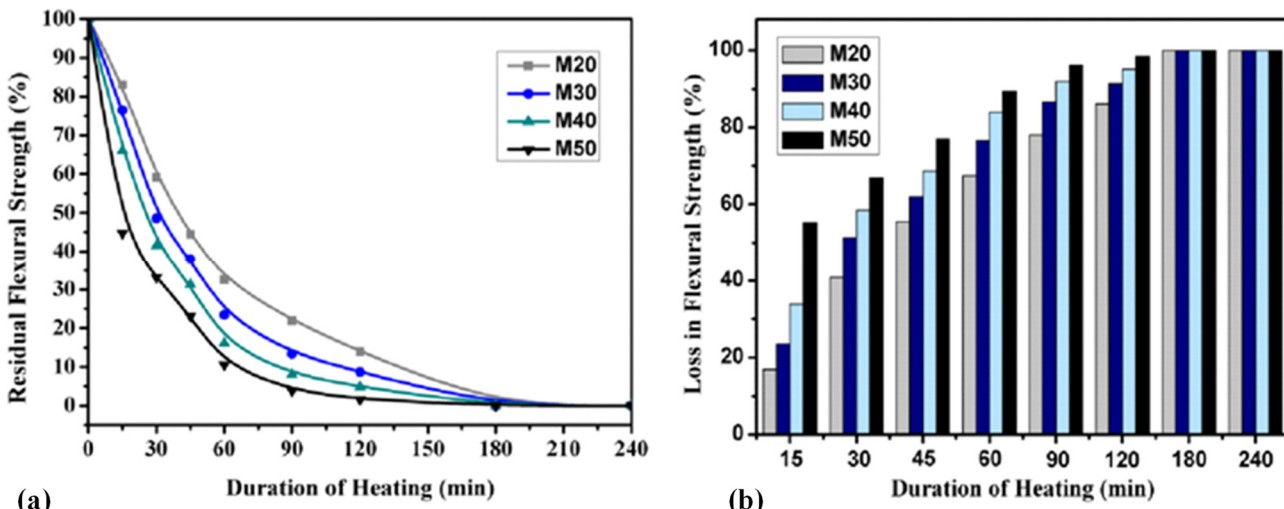

**Figure 8.** (**a**) Residual FS of concrete specimens, (**b**) loss of FS of concrete specimens.

*3.7. Relationship between Characteristic Compressive Strength and Residual Compressive Strength of Concrete*

A relationship was established between the characteristic compressive strength and residual compressive strength of concrete at various heating durations. Let $fck_{(t)}$ be the residual CS and fck be the characteristic CS. Let $fck_{(t)}$ be the residual tensile strength and $fcr_{(t)}$ be the residual flexural strength. A regression analysis was performed to determine the relationship between the parameters described at various heating durations. The proposed equations rely on the concrete grade and applied heating duration as per ISO 834. These equations were applied to strength grades of M-20 to M-50. The empirical relations showed an excellent correlation between the predicted and experimental strength results, and the error percentage was smaller. The developed relations were applied to obtain the mechanical strength characteristics of the concrete grades of M20-M50 exposed to 60–240 min of heating. The developed empirical relations are shown in Table 3.

**Table 3.** Empirical relations used to estimate the residual strength of concrete.

| | Heating Duration | Expression | | R-Squared Value |
|---|---|---|---|---|
| Residual CS, $fck_{(t)}$ | 60 min | $0.6395\,fck + 9.81$ | | 0.99 |
| | 120 min | $0.145\,fck + 12.98$ | $20 \leq fck \leq 50$ | 0.97 |
| | 180 min | $0.0805\,fck + 8.50$ | | 0.97 |
| | 240 min | $0.069\,fck + 3.023$ | | 0.99 |
| Residual TS, $fct_{(t)}$ | 60 min | $0.0136\,fck + 0.899$ | | 0.98 |
| | 120 min | $0.0121\,fck + 0.524$ | $20 \leq fck \leq 50$ | 0.99 |
| | 180 min | $0.0056\,fck + 0.404$ | | 0.99 |
| | 240 min | $0.005\,fck + 0.43$ | | 1 |
| Residual FS, $fcr_{(t)}$ | 60 min | $0.126\,fck + 2.899$ | | 0.97 |
| | 120 min | $0.251\,fck + 1.024$ | $20 \leq fck \leq 50$ | 0.98 |
| | 180 min | $0.0856\,fck + 5.404$ | | 1 |
| | 240 min | $0.075\,fck + 0.55$ | | 0.99 |

*3.8. Effect of the Grade of Concrete and Duration of Heating on Weight Loss*

Two distinct gradient patterns exist in the scattered data of Figure 9. The initial gradient was noted from 0 to 60 min of heating, which was continued by the flatter gradient

over 60 min of exposure. Due to the evaporation of chemically bound water, there was an initial weight loss (WL). Because of the degradation of Ca(OH)$_2$, CSH, and CH hydrate, WL gradually increased after 60 min (925 °C) of heating. The water evaporation process began at the initial time duration. Hence, the WL of the specimen that was heated up to 15 min (718 °C) was negligible. After increasing the heating duration to 240 min (1133 °C), the WL values of M-20 and M-50 concrete specimens were 9.56% and 13.16%, respectively. The WL values of M-20 and M-50 concrete specimens were 2.44% and 1.24%, respectively. The influence of concrete-strength grade on WL was difficult to predict. There were two ranges of WL based on the concrete grade. The permeability and w/c ratio of the concrete were the prime factors that directly relied on the WL. The WL of M20 -grade concrete seemed to be the maximum value, i.e., about 5.47% up to 60 min. This might be attributed to the presence of the high water content in the mix; these water molecules tend to evaporate when exposed to elevated temperature causing a higher number of voids. Moreover, for M-50 concrete, it was 4.55%. When the heating duration was increased beyond 60 min, it was observed that the WL was greater for the M-50 mix; this might be attributed to the presence of a lower w/c ratio, which is less porous and has a higher amount of cement paste. The higher amount of cement produced a greater decomposition of Ca(OH)$_2$, CSH, and CH hydrates.

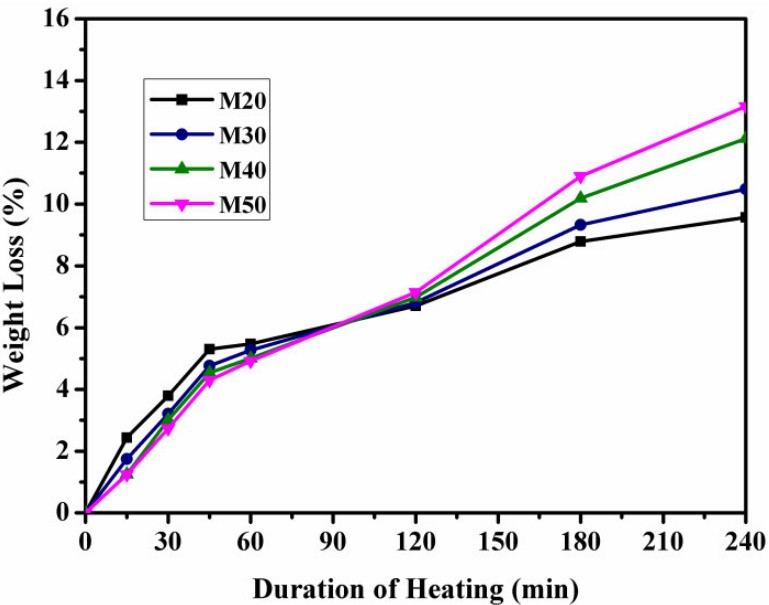

**Figure 9.** WL of concrete specimens subjected to heating.

### 3.9. Surface Spalling Characteristics of Concrete

Surface spalling is the expectable critical observation in concrete subjected to elevated temperatures. The term spalling represents the stage when the separation of concrete occurs after exposure to higher temperatures due to fire loads. Spalling occurs in two stages: the creation of thermal pressure in the concrete pore structure and the shrinkage of the concrete. From the detailed assessment of the experimental results, it was observed that the surface spalling of concrete was also included based on the w/c ratio and intensity of temperature exposure, respectively. The before (0 min) and after exposure (60, 120, and 240 min) surface changes of M-20 and M-50 concrete samples were examined, and are presented in Figure 10. From the surface changes in concrete, following exposure for 60 min, both M-20 and M-50 samples showed small cracks. After 120 min of exposure, the cracks increased and voids were formed. The M-20 and M-50 specimens developed extensive surface cracks, surface spalling, and large voids. Due to the high temperature and sudden forced cooling, large cracks were created on the hot surfaces, and this led to strength reduction by the decomposition of the CHS gel layer.

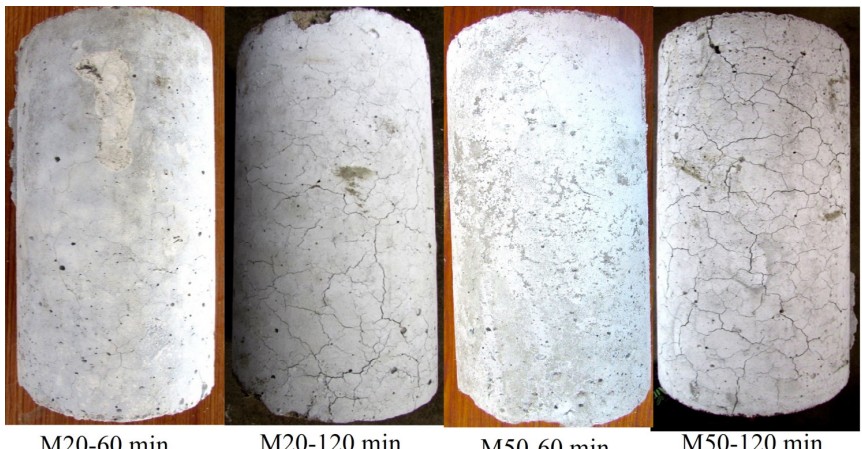

| M20-60 min | M20-120 min | M50-60 min | M50-120 min |

**Figure 10.** Surface change in concrete specimens exposed to elevated temperatures.

*3.10. Crack-Width Measurement of Concrete*

Detailed information on the crack-width measurement of concrete has also been discussed in the study, and it must be noted that it has not been presented in past studies. Elcometer equipment was used in the study to determine the crack width in the concrete. The range of the elcometer varies between 0.0 to 1.8 mm, with the lowest count of 0.02 mm. After exposure to the ISO standard curve and forced water cooling, the average crack-width values of the cube, cylinder, and beam specimens were noted. The measured crack-width images of M-20 to M-50 are illustrated in Figure 11. From the visual observations of the concrete, it can be observed that the grade of concrete and duration of heating widened the thermal cracks, and the number of surface cracks on the concrete also gradually increased.

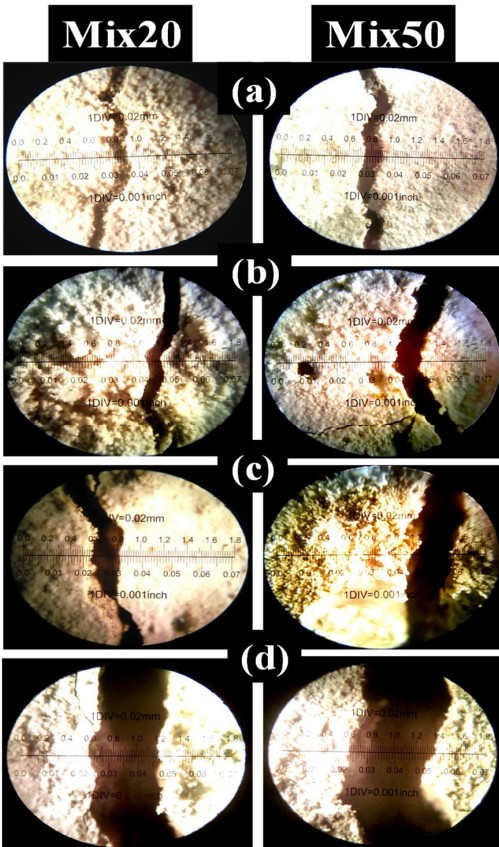

**Figure 11.** Crack width of concrete specimens exposed to elevated temperatures: (**a**) 30, (**b**) 60 min, (**c**) 90, and (**d**) 120 min.

Figure 12 shows the measured crack-width values of concrete specimens after sudden cooling. From the results of the crack-width analysis, it was ascertained that with an increase in the duration of heating, the crack-width values of all the grades of concrete were observed to significantly increase. The thermal crack width of M-20 grade was 0.05–0.53 mm. For M-30 and M-40 samples, the noted range was between 0.08–0.65 and 0.11–0.72 mm. M-50 concrete's thermal crack width varied between 0.12–0.82 mm. Between 15 and 60 min of exposure, the thermal crack-width values were less. Therefore, higher crack-width values were observed when heat exposure occurred between 60 and 240 min. The effect of elevated temperature caused severe damage to higher-grade concrete.

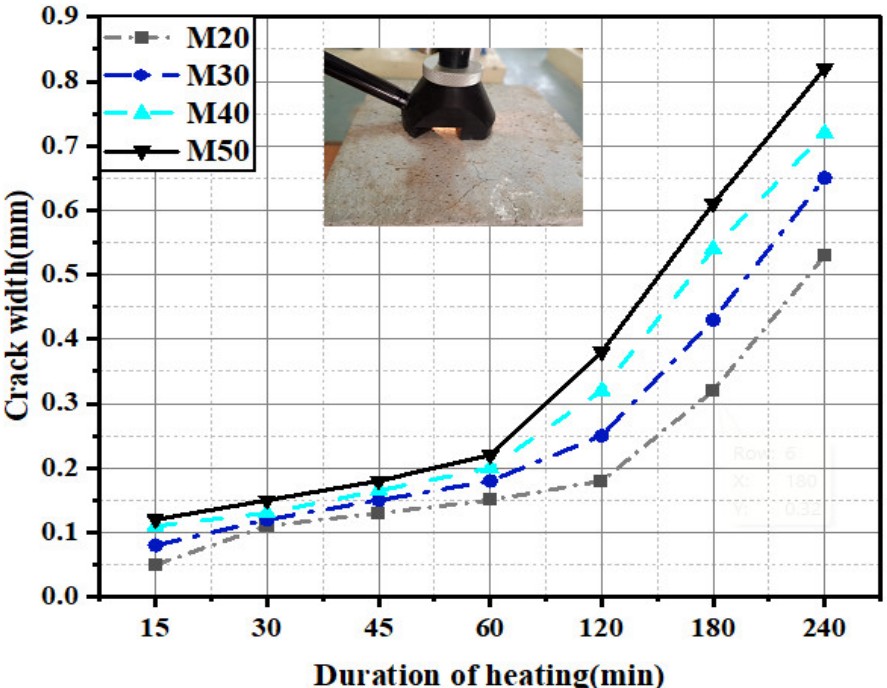

**Figure 12.** Microscopic view and range of crack widths in M-20 to M-50 concrete specimens after prolonged temperature exposure.

## 4. SEM Analysis

Scanning Electron Microscope (SEM) analysis was performed for reference and heated (240 min) concrete M-20 and M-50 specimens. The analysis was conducted to observe the microcracks, voids, and dispersion of CSH gel before and after temperature exposure. The SEM concrete samples were collected from the damaged specimens with a dimension of $0.24 \times 0.24 \times 0.1$ mm. The SEM images depicted the cracks and porosity after exposure to higher temperatures. Figure 13 shows the microstructure images of reference and heated M-20- and M-50-grade concrete specimens. The analysis showed that the reference M-20 and M-50 concrete specimens with a well-packed, dense structure and well-hydrated C-H and CSH gel presented no pores and voids. In the M-50 specimen with more CSH gel and dense structure, more pores and voids were observed than in the M-20 specimens. At 60 min of exposure, the M20 concrete with well-packed hydrated CSH gel showed minor cracks. The M-50 specimen exhibited minor cracks and bond loss between aggregate and CSH gel. At prolonged temperatures (240 min), the M-20 heat-exposed specimen exhibited cracks with the dehydrated CSH phase. However, in the case of the M-50 specimen, large cracks with greater voids were observed. This was due to the high temperatures that deteriorated the bond between $Ca(OH)_2$ and the CSH gel, which led to greater strength loss of M-50 than the M-20 concrete. Similar microstructure changes in the concrete led to greater strength loss for both lower and higher grades of concrete stated by [1,28]. Due to these changes, the higher grade of concrete revealed greater strength loss at high-temperature exposure.

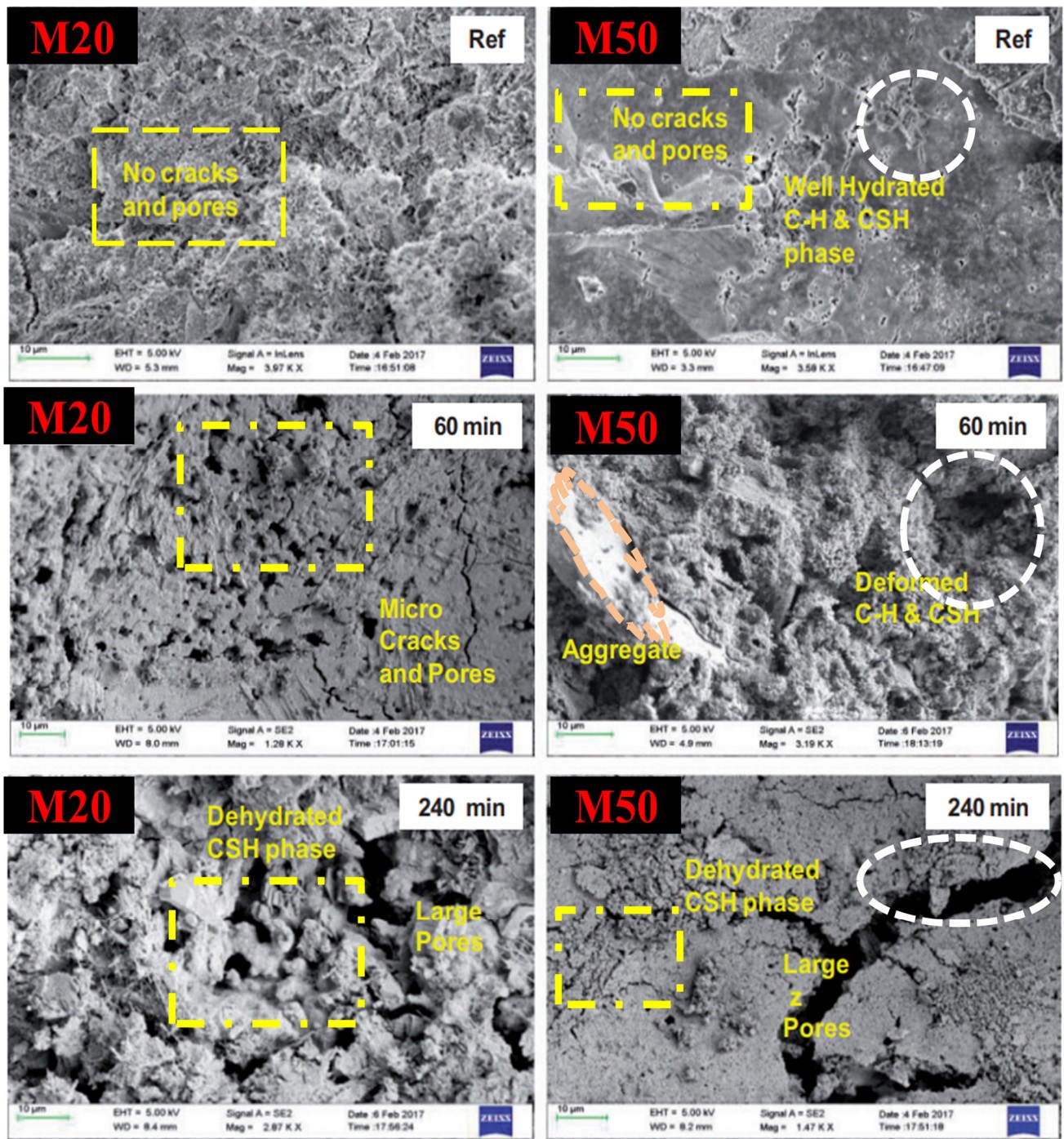

**Figure 13.** SEM images of M-20 and M-50 concrete exposed to high temperatures (Reference, 60 and 240 min).

## 5. Image Analysis

The implementation of damage detection through fire-affected concrete images was conducted to effectively determine the extent of the damage. The image analysis implementation was executed in a MATLAB environment. The images were obtained from real-time exposure experiments with constant illumination. These images were collected from different heating durations, such as 30, 60, 90, and 120 min. Two major steps were followed to detect the damage presented in the images. The input images were diagnosed using an anisotropic filter [46]. The diagnosed image was used to find the damage in the concrete images. The technique adopted by [47] was used to highlight the cracks of

the fire-affected concrete image. The ripplet transform, a higher-order wavelet transform, was used to effectively engrave the damages. It was widely used to understand images and medical image analysis [48]. Figure 14 shows a few sample images of the inputs and their damage.

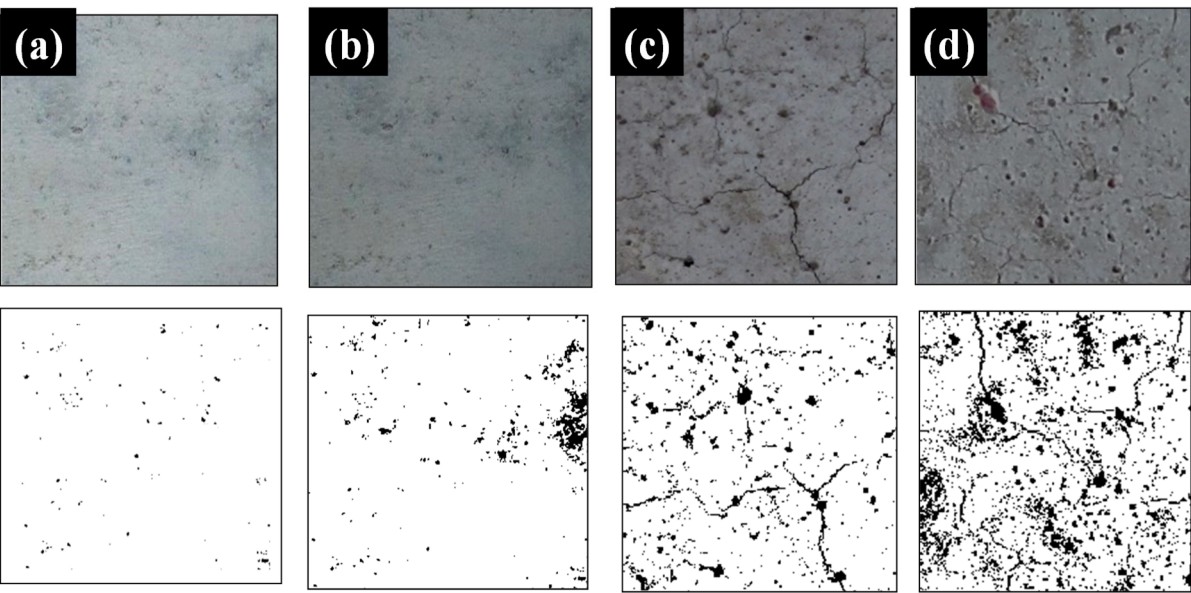

**Figure 14.** Image-based damage detection: (**a**) 30, (**b**) 60, (**c**) 90, (**d**) 120 min.

The software-based implementation yielded a more accurate result. The quantification of damages was performed after the damage-detection process. The damage was quantified through the crack-length parameter. The method adopted by [47] was used to quantify the cracks in pixels. When the heating duration increased, the damage occurrence was greater. This was observed by measuring the total crack length. The average crack lengths of the specimen heated up to 30, 60, 90, and 120 min were as follows: 167, 1290, 9894, and 10,120 pixels. As the thermal exposure increased, the crack propagation also increased, and this induced greater damage to the concrete. This was analyzed through image processing-based experiments.

## 6. Relevance of the Proposed Research for Practical Implementation

Fire poses a significant threat to life, even during seemingly safe situations. Every minute is critical when evacuating inhabitants when a fire breaks out in a building. Conducting experiments on the fire-resistance capacity of materials is too risky and costly. Therefore, the residual strength data on concrete are helpful in developing different applications for safety and security. Experimental data extracted from investigations on strength capacity may be useful for the analytical modeling of modified structural members under high temperatures as an input to the finite element model. Material strength values are also essential for estimating the strength of reinforced members subjected to various heating durations. This enables the adoption of a suitable repair techniques to improve the durability of fire-affected buildings. It is useful to identify the appropriate grades of concrete for fire-endurance purposes, as it is an important design parameter. Furthermore, developing an empirical relationship between residual strengths is beneficial for design applications. The data on residual strength presents the issue of safe materials used during fire exposure, which will provide a guideline for preparing a preliminary design for civil/structural engineers.

## 7. Conclusions

A detailed investigation was conducted to evaluate the residual strength properties of different grades of concrete mixes after being cooled by water. The strength properties, such as CS, TS, and FS, were examined. Physical characteristics and morphological changes were analyzed. Based on the experimental investigations conducted on the different grades of concrete and the effect of elevated temperatures, the following conclusions were drawn:

○ As the duration of heating increased, the UPV values of heated concrete specimens decreased for all the grades of concrete. The UPV result shows the poor quality of concrete after 60 min of exposure and zero values at 180 and 240 min of exposure. At 45 min, the UPV values of M-20 and M30 specimens showed poor quality, whereas M-40 and M50 specimens exhibited good quality.

○ After exposure for a prolonged duration beyond 60 min, concrete specimens with a higher grade (M-50) exhibited a higher rate of strength or weight loss compared to the lower grade (M-20).

○ The loss of compressive strength, tensile strength, and flexural strength ranged from 22–30%, 54–69%, and 70–88%, respectively, for M20–M50 concrete specimens exposed to 60 min duration of heating.

○ The loss of compressive strength, tensile strength, and flexural strength ranged from 51–65%, 70–77%, and 85–97%, respectively, for M20–M50 concrete specimens exposed to 120 min duration of heating. The flexural strength was susceptible to higher temperatures.

○ After 120 min of heating, the residual flexural strength of concrete decreased to 0% for all the grades of concrete, and the compressive and tensile strength values were greater than that of the flexural strength of concrete.

○ The influence of the water–cement ratio played a key role in the morphological changes of concrete. While the heating duration increased, the surface spalling, crack width, and weight loss of concrete gradually increased. Additionally, the damage evaluation was confirmed with the SEM and image analysis.

○ Weight loss was found to increase along with the heating duration and strength grade of concrete. For the concrete specimens exposed to 240 min of heating, the weight loss values of M-20 and M-50 concrete samples were 10.36% and 13.16%, respectively.

○ Overall, the higher water–cement-ratio concrete performed better than lower water–cement-ratio concrete after exposure to elevated temperatures.

○ A simple empirical relationship was developed using regression analysis to determine the residual strength by varying the duration of heating and grade of concrete.

**Author Contributions:** D.P.T., B.K., T.K., A.N. and A.D.A.: investigation, methodology, formal analysis, writing—original draft, writing—review and editing. D.P.T. and A.N.: resources, supervised the research, and analyzed the results. A.N., D.P.T., B.K. and T.K.: project administration, visualization, and review and editing. A.N., B.G.A.G. and K.R.: validation, suggested and chose the journal for submission. A.N., B.G.A.G. and K.R.: collaborated, coordinated the research, and reviewed it for submission. All authors have read and agreed to the published version of the manuscript.

**Funding:** This research received no external funding.

**Institutional Review Board Statement:** Not applicable.

**Informed Consent Statement:** Not applicable.

**Data Availability Statement:** The data presented in this study are available on request from the corresponding author.

**Conflicts of Interest:** This manuscript has neither been submitted to, nor is under review by another journal or other publishing venue. The authors have no affiliations with any organization with a direct or indirect financial interest in the subject matter discussed in the manuscript. The authors declare no conflict of interest.

## Abbreviations

| | |
|---|---|
| CS | Compressive Strength |
| TS | Tensile Strength |
| FS | Flexural Strength |
| CA | Coarse Aggregate |
| WL | Weight Loss |
| CSH | Calcium Silicate Hydrate |
| $Ca(OH)_2$ | Calcium Hydroxide |
| w/c ratio | Water-to-Cement Ratio |
| M-20 | Concrete with Strength 20 MPa |
| M-30 | Concrete with Strength 30 MPa |
| M-40 | Concrete with Strength 40 MPa |
| M-50 | Concrete with Strength 50 MPa |
| NSC | Normal-Strength Concrete |
| HSC | High-Strength Concrete |
| *fck* | Characteristic Compressive Strength of Concrete (MPa) |
| *fck*$_{(t)}$ | Residual Compressive Strength |
| *fct*$_{(t)}$ | Residual Tensile Strength |
| *fcr*$_{(t)}$ | Residual Flexural Strength |
| C-H | Calcium Hydroxide |

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
