# Peer review of "Influence of Heating–Cooling Regime on the Engineering Properties of Structural Concrete Subjected to Elevated Temperature"

_buildings, doi:10.3390/buildings13020295_

Round 1
Reviewer 1 Report
The conducted work “Influence of heating cooling regime on engineering properties of structural concrete subjected to higher temperature exposure” is good. However, following comments should be addressed to further improve the paper:
1. Explicitly mention the novelty and research significance of current work in last paragraph of introduction section with emphasis on scientific soundness. The sentence “experiments on the fast cooling (water cooling) effect on physical, mechanical, and microstructure properties are minimal.” is little vague, what is being learnt from these minimal researches?
2. Avoid paragraph of few (2-4) sentences throughout the manuscript, particularly in results and discussions sections e.g. lines 33-37, 38-44, etc.
3. Show standard deviation where-ever the average is being taken.
4. Results should be further elaborated with scientific reasoning.
5. A separate brief section (explaining the relevance of this research for practical implementation) may be added before conclusion section.
6. Conclusions should be reflection of obtained results with scientific soundness. Conclusions are little long; these should be to the point as obtained from results. Closing remarks should be added at the end of conclusion section keeping in mind all conclusive bullet points.
7. English Language should be improved throughout the manuscript.
B. SPECIFIC COMMENTS FOR IMPROVING FOCUSSED RESEARCH
1. The word “test” from caption of heading 3.1 should be deleted as this heading talks about results. Similarly, procedures of all tests being performed should be briefly explained in section 2.
2. The test procedure of results presented in section 3.1 should be explained in section 2.
3. Lines 454-455: how much loss is observed and what could be the reason?
4. Lines 456-457: how much increase is there and what could be the reason? The word “significantly” seems little vague in conclusion. Similarly, remaining conclusions need major rephrasing with scientific soundness.
Author Response
Responses to Reviewer comments
The authors sincerely thank the editor and reviewer 1 for the constructive comments. All of the comments have been carefully considered and, wherever appropriate, revisions have been made to the manuscript. Responses to these comments and revisions implemented in the paper are detailed below. The Reviewer comments are in italic black fonts while our replies are in blue. The text changed in the manuscript is highlighted in red.
Reviewer # 1
Comments and Suggestions for Authors
The conducted work “Influence of heating cooling regime on engineering properties of structural concrete subjected to higher temperature exposure” is good. However, following the comments should be addressed to further improve the paper:
(Q1) Explicitly mention the novelty and research significance of current work in the last paragraph of in the introduction section with emphasis on scientific soundness. The sentence "experiments on the fast cooling (water cooling) effect on physical, mechanical, and microstructure properties are minimal." is a little vague, what is being learnt from these minimal researches?
(A1) We thank reviewer 1 for this comment. As per the reviewers’ suggestion, we have stated this research paper's novelty and research significance in detail, as given below:
The primary reason for the damage level of fire-affected concrete depends on the magnitude of temperature, concrete mix, and cooling type adopted. During fire accidents, generally, concrete structures are quenched by water extinguishers. The damage level of water-cooled concrete structures is vital to evaluate the residual strength for repair and rehabilitation works.
Research on fire-damaged concrete gives the knowledge of residual strength and the behaviour of structural parts following the fire event. The structural, thermal, and material properties of concrete subjected to high temperatures and cooled by water are not available in the literature. In this study, the concrete specimens were heated according to the ISO 834 fire rating. The mechanical performance of concrete specimens was determined when subjected to high temperatures for different durations.
When concrete is exposed to elevated temperature, its internal microstructure changes, and thereby strength and durability of concrete decrease. The research aims to investigate the influence of elevated temperature on the strength and microstructure characteristics of concrete with different strength grades. The structural and thermal behaviour of the members subjected to temperature loads will give an overview of how they respond to elevated temperatures. The selection of good quality fire-resistant materials is now getting due attention. The kind of materials used in construction and how they respond to fire greatly affect how a structure responds to fire. It is, therefore, essential for understanding the behaviour of structural materials subjected to fire and their mechanical response to heating. Since the responsibility of fire-resistant design is delegated to the structural engineers, they must have an idea about the behaviour of materials under fire. With the increased incidents of major building fires, assessment, repairs, and rehabilitation of fire-damaged structures have become a topic of interest. Structural concrete's post-fire response and fire-retardant capacity are essential parameters much needed for field practices.
The data on the physical, mechanical, and microstructure properties of water-cooled specimens may be relevant in predicting the evacuation time of humans during building fires. Since the residual hardening performance for water-cooled specimens differs significantly from air-cooled specimens, this data will be helpful in formulating structural fire standards. There is a scarcity of fundamental knowledge on the influence of standard fire on the thermal characteristics of concrete, notably for concrete with varying strength grades. While knowledge and experience with various types of concrete behaviour at ambient and increased temperatures are well established, the effect of water cooling on concrete with varied grades of concrete subjected to typical fire requires further investigation. The study aims to set a database for the residual hardening performance of concrete subjected to a standard fire and to anticipate the mechanical behaviour of concrete at increased temperatures and exposed to fast cooling circumstances.
Investigation of different mechanical and microstructure properties of concrete exposed to fire cooled by quenching water by various researchers:
Researchers have investigated different mechanical and microstructure properties of concrete exposed to fire [1-3]. These studies focus mainly on the effect of high temperatures on the strength of the concrete. Concrete is an important construction material. In reinforced concrete, steel is protected by concrete from elevated temperatures. The behaviour of concrete under fire needs to be investigated to ensure the safety of structures and occupants during fire accidents [1].
Moreover, most of the research data on residual properties of concrete after the fire are available under the conditions of natural air cooling. The residual strength during and after the fire differs in structural elements. However, it depends on the cooling regimes, i.e., gradual natural air cooling or forced water cooling. Water spraying is the most commonly adopted method of extinguishing a fire. Water cooling causes a thermal shock [4] and results in a severe reduction in the mechanical properties of concrete. Therefore, the cooling regimes significantly influence the residual mechanical properties of concrete [5-8].
It has been reported that water cooling caused a more severe decrease in strength compared to natural cooling [9-11]. Therefore, the effect of cooling regimes on the mechanical properties of concrete is of great concern, especially after a fire case was reported [4, 12]. Many authors have investigated the effect of different cooling methods on the strength of concrete. In general, it is observed that a short immersion of heated concrete specimens in water has not significantly affected the reduction of concrete strength [13]. It is reported that the specimens cooled naturally in the air lost only about 10% of their strength. Specimens cooled rapidly in water for a short time of 5 minutes and lost about 35% of their strength. Specimens cooled rapidly in water for a long time of 20 minutes lost about 55% of their strength [14].
Normally the results of strength loss due to spraying occur between the reduction obtained through natural cooling and the reduction obtained through quenching [15]. In general, it is noticed that faster cooling rates result in an additional strength loss compared to the naturally slow cooling process. This additional strength reduction is attributed to a thermal shock, which results in a sudden temperature difference [4].
References
- Sarshar, R., & Khoury, G. A. (1993). Material and environmental factors influencing the compressive strength of unsealed cement paste and concrete at high temperatures. Magazine of concrete research, 45(162), 51-61.
- Poon, C. S., Azhar, S., Anson, M., & Wong, Y. L. (2001). Strength and durability recovery of fire-damaged concrete after post-fire-curing. Cement and concrete research, 31(9), 1307-1318
- Annerel, E. (2010). Assessment of the Residual Strength of Concrete Structures after Fire Exposure (Doctoral dissertation, Ghent University).
- Peng, G. F., Bian, S. H., Guo, Z. Q., Zhao, J., Peng, X. L., & Jiang, Y. C. (2008). Effect of thermal shock due to rapid cooling on residual mechanical properties of fiber concrete exposed to high temperatures. Construction and Building Materials, 22(5), 948-955.
- Phan, L. T., & Phan, L. T. (1996). Fire performance of high-strength concrete: A report of the state-of-the-art. Gaithersburg, MD: US Department of Commerce, Technology Administration, National Institute of Standards and Technology, Office of Applied Economics, Building and Fire Research Laboratory.
- Schneider, U. (1988). Concrete at high temperatures—a general review. Fire safety journal, 13(1), 55-68.
- Chan, Y. N., Peng, G. F., & Anson, M. (1999). Residual strength and pore structure of high-strength concrete and normal strength concrete after exposure to high temperatures. Cement and concrete composites, 21(1), 23-27.
- Khoury, G. A. (1992). Compressive strength of concrete at high temperatures: a reassessment. Magazine of Concrete Research, 44(161), 291-309.
- Luo, X., Sun, W., & Chan, S. Y. N. (2000). Effect of heating and cooling regimes on residual strength and microstructure of normal strength and high-performance concrete. Cement and Concrete Research, 30(3), 379-383.
- Ishihara, S., Goshima, T., Nomura, K., & Yoshimoto, T. (1999). Crack propagation behavior of cermets and cemented carbides under repeated thermal shocks by the improved quench test. Journal of materials science, 34(3), 629-636.
- Nassif, A. Y. (2002). Post firing stress‐strain hysteresis of concrete subjected to various heating and cooling regimes. Fire and materials, 26(3), 103-109.
- Anand, N., & Godwin, A. (2016). Influence of mineral admixtures on mechanical properties of self‐compacting concrete under elevated temperature. Fire and Materials, 7(40), 940-958.
- Abramowicz, M., & Kowalski, R. (2005). The influence of short time water cooling on the mechanical properties of concrete heated up to high temperature. Journal of civil engineering and management, 11(2), 85-90.
- Kowalski, R. (2007). The effects of the cooling rate on the residual properties of heated-up concrete. Structural concrete, 8(1), 11-15.
- Botte, W., & Caspeele, R. (2017). Post-cooling properties of concrete exposed to fire. Fire Safety Journal, 92, 142-150.
(Q2) Avoid paragraphs of a few (2-4) sentences throughout the manuscript, particularly in results and discussions sections e.g., lines 33-37, 38-44, etc.
(A2) We thank reviewer 1 for this comment. As per your comment, a few sentences have now been deleted from the revised manuscript.
(Q3) Show standard deviation wherever the average is being taken.
(A3) We thank reviewer 1 for the valuable comment. As per the suggestion, the standard deviation is shown for M20 and M50 concrete grades. The observed standard deviations of compressive strength, tensile strength, and flexural strength are 0.59, 0.11, and 0.26, respectively for M20 concrete. For M50 concrete it was 0.84, 0.07, and 0.13, respectively.
|
|
M20 (Compressive Strength) |
M50 (Compressive Strength) |
||||||||
|
|
S1 |
S2 |
S3 |
Mean |
SD |
S1 |
S2 |
S3 |
Mean |
SD |
|
0 |
24.1 |
23.01 |
24.6 |
23.90 |
0.81 |
56.02 |
57.12 |
56.89 |
56.68 |
0.58 |
|
15 |
17.9 |
18.28 |
18.32 |
18.17 |
0.23 |
50.47 |
51.65 |
49.88 |
50.67 |
0.90 |
|
30 |
22.27 |
21.98 |
22.01 |
22.09 |
0.16 |
58.81 |
58.32 |
56.75 |
57.96 |
1.08 |
|
45 |
22.8 |
19.88 |
19.65 |
20.78 |
1.76 |
46.1 |
47.25 |
45.66 |
46.34 |
0.82 |
|
60 |
18.9 |
18.98 |
17.95 |
18.61 |
0.57 |
39.9 |
40.78 |
38.48 |
39.72 |
1.16 |
|
120 |
11.16 |
12.1 |
11.89 |
11.72 |
0.49 |
18.53 |
20.24 |
19.89 |
19.55 |
0.90 |
|
180 |
6.2 |
7.21 |
6.89 |
6.77 |
0.52 |
10.16 |
11.39 |
9.89 |
10.48 |
0.80 |
|
240 |
3.22 |
3.5 |
3.4 |
3.37 |
0.14 |
4.59 |
5.5 |
4.78 |
4.96 |
0.48 |
|
|
M20 (Tensile Strength) |
M50 (Tensile Strength) |
||||||||
|
|
S1 |
S2 |
S3 |
Mean |
SD |
S1 |
S2 |
S3 |
Mean |
SD |
|
0 |
2.62 |
2.32 |
2.51 |
2.48 |
0.15 |
4.87 |
5.09 |
5.1 |
5.02 |
0.13 |
|
15 |
2.05 |
1.92 |
1.98 |
1.98 |
0.07 |
3.5 |
3.42 |
3.39 |
3.44 |
0.06 |
|
30 |
1.59 |
1.89 |
1.69 |
1.72 |
0.15 |
2.72 |
2.65 |
2.81 |
2.73 |
0.08 |
|
45 |
1.4 |
1.55 |
1.42 |
1.46 |
0.08 |
1.9 |
1.85 |
2.02 |
1.92 |
0.09 |
|
60 |
1.01 |
1.21 |
1.19 |
1.14 |
0.11 |
1.48 |
1.65 |
1.49 |
1.54 |
0.10 |
|
120 |
0.78 |
0.89 |
0.51 |
0.73 |
0.20 |
1.11 |
1.08 |
1.15 |
1.11 |
0.04 |
|
180 |
0.39 |
0.56 |
0.51 |
0.49 |
0.09 |
0.59 |
0.65 |
0.69 |
0.64 |
0.05 |
|
240 |
0.26 |
0.29 |
0.24 |
0.26 |
0.03 |
0.42 |
0.38 |
0.35 |
0.38 |
0.04 |
|
|
M20 (Flexural Strength) |
M50 (Flexural Strength) |
||||||||
|
|
S1 |
S2 |
S3 |
Mean |
SD |
S1 |
S2 |
S3 |
Mean |
SD |
|
0 |
2.22 |
3.69 |
3.12 |
3.01 |
0.74 |
5.90 |
6.62 |
5.98 |
6.17 |
0.39 |
|
15 |
2.12 |
3.20 |
2.19 |
2.50 |
0.60 |
2.65 |
2.98 |
2.65 |
2.76 |
0.19 |
|
30 |
1.48 |
2.18 |
1.69 |
1.78 |
0.36 |
2.00 |
2.15 |
1.98 |
2.04 |
0.09 |
|
45 |
1.23 |
1.05 |
1.42 |
1.23 |
0.19 |
1.22 |
1.50 |
1.12 |
1.28 |
0.20 |
|
60 |
1.05 |
0.78 |
0.88 |
0.90 |
0.14 |
0.61 |
0.68 |
0.89 |
0.73 |
0.15 |
|
120 |
0.44 |
0.39 |
0.51 |
0.45 |
0.06 |
0.15 |
0.18 |
0.15 |
0.16 |
0.02 |
|
180 |
0.00 |
0.00 |
0.00 |
0.00 |
0.00 |
0.00 |
0.00 |
0.00 |
0.00 |
0.00 |
|
240 |
0.00 |
0.00 |
0.00 |
0.00 |
0.00 |
0.00 |
0.00 |
0.00 |
0.00 |
0.00 |
(Q4) Results should be further elaborated with scientific reasoning.
(A4) We thank reviewer 1 for the valuable comment. The results are elaborated with scientific reasoning and updated in the manuscript. The given discussions are updated in the appropriate section of the manuscript.
Elaborated results with scientific reasoning:
Beyond 30 minutes, a higher reduction occurred in M-50 concrete specimens than in other specimens. This drop in strength is due to dehydration and decomposition of internal chemical structure in cement paste as the exposure duration and temperature increase (Chan et al., 1999; Lin et al., 1996). The reduction in strength is more pronounced based on moisture content, density, heating rate, and porosity. The weakening of cement paste causes the disintegration of concrete. Hydrated cement paste contains CSH, CH, and ettringite. At 100°C, normally free water and absorbed water of the cement paste gets evaporated (Savva et al., 2005). A desiccation occurs in this state, which is the reason for cracks in the surface. But above 400°C, chemically combined water also started losing at this state. The literature has reported that the strength loss is largely attributed to the decomposition of calcium hydroxide, which generally occurs over 500°C (Khaliq and Kodur, 2011).
From 180 to 240 minutes of exposure, the effect of heating duration and temperature surpasses the effect of CS of concrete on the strength loss of specimens. Normally above 500°C, dehydration and disintegration of CSH gel occur. Also, remarkable chemical changes take place in the cement paste. Generally, over 900°C, a complete decomposition takes place in the concrete, and thereby concrete losses its strength, as reported by Topcu et al. (2011). The reduction in strength is due to the dehydration of absorbed water and chemically bound water (Arioz, 2007). The decrease in strength may be due to decomposition, deterioration, and thermal incompatibility, as reported by Xiong and Liew (2015).
The reduction in TS was higher for specimens with higher grades than lower grades at all heating durations, and an additional strength loss was noticed. It has been speculated (Fares et al., 2009; Kalifa et al., 2000; Phan et al., 2001; Bastami et al., 2011; Sideris et al., 2009) that the higher density in high-strength concrete restricts the water vapour pressure from escaping. This internal pore pressure exceeds the tensile strength of concrete, which results in large numbers of micro and macro cracks (Sideris, 2007; Chan et al., 1999; Khoury, 1992).
So, crack formation under flexure was found to be more critical than compression. An increase in temperature exposure increases crack propagation; this might be attributed to the degradation of CSH gel phases (Correia et al.,2014). Heated specimens are more prone to crack under flexural loading and close up under compressive loading. Thus, the impact of crack coalescence is more crucial on the flexural strength than that of the compressive strength of concrete specimens (Potha Raju et al., 2004; Aydın and Baradan, 2007).
References:
Chan, Y.N., Peng, G.F. and Anson, M. (1999), “Residual strength and pore structure of high-strength concrete and normal strength concrete after exposure to high temperatures”, Cement and Concrete Composites, Vol. 21 No. 1, pp. 23-27.
Lin, W.M., Lin, T.D. and Powers-Couche, L.J. (1996), “Microstructures of fire-damaged concrete”, Materials Journal, Vol. 93 No. 3, pp. 199-205.
Savva, A., Manita, P. and Sideris, K.K. (2005), “Influence of elevated temperatures on the mechanical properties of blended cement concretes prepared with limestone and siliceous aggregates”, Cement and Concrete Composites, Vol. 27 No. 2, pp. 239-248.
Khaliq, W. and Kodur, V. (2011), “Thermal and mechanical properties of fiber reinforced high performance self-consolidating concrete at elevated temperatures”, Cement and Concrete Research, Vol. 41 No. 11, pp. 1112-1122.
Topcu, I.B., Boga, AR and Demir, A. (2011), “Influence of cover thickness on the mechanical properties of steel bar in mortar exposed to high temperatures”, Fire and Materials, Vol. 35 No. 2, pp. 93-103.
Arioz, O. (2007), “Effects of elevated temperatures on properties of concrete”, Fire safety journal, Vol. 42 No. 8, pp. 516-522.
Xiong, M.X. and Liew, J.R. (2015), “Spalling behavior and residual resistance of fibre reinforced Ultra-High performance concrete after exposure to high temperatures”, Materiales de Construccion, Vol. 65 No. 320, pp. 071.
Fares, H., Noumowe, A. and Remond, S. (2009), “Self-consolidating concrete subjected to high temperature: mechanical and physicochemical properties”, Cement and Concrete Research, Vol. 39 No. 12, pp. 1230-1238.
Kalifa, P., Menneteau, F.D. and Quenard, D. (2000), “Spalling and pore pressure in HPC at high temperatures”, Cement and concrete research, Vol. 30 No. 12, pp. 1915-1927.
Phan, L.T., Lawson, J.R. and Davis, F.L. (2001), “Effects of elevated temperature exposure on heating characteristics, spalling, and residual properties of high performance concrete”, Materials and Structures, Vol. 34 No. 2, pp. 83-91.
Bastami, M., Chaboki-Khiabani, A., Baghbadrani, M. and Kordi, M. (2011), “Performance of high strength concretes at elevated temperatures”, Scientia Iranica, vol. 18 No. 5, pp. 1028-1036.
Sideris, K.K., Manita, P. and Chaniotakis, E. (2009), “Performance of thermally damaged fibre reinforced concretes”, Construction and Building Materials, Vol. 23 No. 3, pp. 1232-1239.
Sideris, K.K. (2007), “Mechanical characteristics of self-consolidating concretes exposed to elevated temperatures”, Journal of materials in civil engineering, Vol. 19 No. 8, pp. 648-654.
Khoury, G.A. (1992), “Compressive strength of concrete at high temperatures: a reassessment”, Magazine of concrete Research, Vol. 44 No. 161, pp. 291-309.
Correia JR, Lima JS and de Brito J. Post-fire mechanical performance of concrete made with selected plastic waste aggregates. Cement and Concrete Composites 2014; 53: 187-199.
Potha Raju M, Shobha M and Rambabu K. Flexural strength of fly ash concrete under elevated temperatures. Magazine of Concrete Research 2004; 56: 83-88.
Aydın S and Baradan B. Effect of pumice and fly ash incorporation on high temperature resistance of cement based mortars. Cement and Concrete Research 2007; 37: 988-995.
(Q5) A separate brief section (explaining the relevance of this research for practical implementation) may be added before the conclusion section.
(A5) We thank reviewer 1 for the valuable comment. A separate brief section explaining the relevance of this research for practical implementation has been added and updated in the manuscript.
Relevance of the proposed research for practical implementation:
Fire poses a significant threat to life safety even under a perfect shelter for humanity. Every minute is critical in evacuating inhabitants when fire breaks out in a building. Conducting experiments on the fire resistance capacity of materials is too risky and costly. Therefore, the residual strength data on concrete will be helpful in developing different applications for safety and security. Experimental data extracted from investigations on strength capacity may be useful for the analytical modelling of modified structural members under elevated temperatures as an input to the finite element model. Material strength values are also essential for estimating the strength of reinforced members subjected to various heating durations. This enables in adoption of a suitable repair technique to improve the durability of fire-affected buildings. It will be useful to identify the appropriate grade of concrete for ample fire endurance, as it is an important design parameter. Further, developing an empirical relationship between residual strengths will become beneficial for design applications. The data on residual strength gives an idea about the reduction in material safety during fire exposure, which will provide the guideline for preparing the preliminary design for civil/structural engineers.
(Q6) Conclusions should be reflection of obtained results with scientific soundness. Conclusions are little long; these should be to the point as obtained from results. Closing remarks should be added at the end of the conclusion section keeping in mind all conclusive bullet points.
(A6) We thank reviewer 1 for the valuable comment. Accordingly, conclusions have been made short and precise with valid mentioning of results and updated in the revised manuscript.
Revised conclusion:
A detailed investigation was carried out to evaluate the residual strength properties of different grades of concrete mixes after being cooled by water. The strength properties such as CS, TS, and FS were examined. The physical characteristics and morphological changes were analysed. Based on the experimental investigations conducted on the grade of concrete and the effect of elevated temperature, the following conclusions were drawn:
- As the duration of heating increased, the UPV values of heated concrete specimens declined for all the grades of concrete. The UPV result showed poor quality in concrete after 60-minute exposure and zero values at 180 and 240-minute exposure. At 45 minutes, the UPV values of M-20 and M30 specimens showed doubtful quality, whereas, M-40 and M50 specimens exhibited good quality.
- After exposure to a prolonged duration of beyond 60 minutes, concrete specimens with higher grade (M50) exhibited a higher rate of strength loss or weight loss compared to lower strength grade (M20).
- The loss in compressive strength, tensile strength, and flexural strength ranged from 22% - 30%, 54% - 69%, and 70% - 88% for M20 - M50 concrete specimens exposed to a 60-minute duration of heating.
- The loss in compressive strength, tensile strength, and flexural strength ranged from 51% - 65%, 70% - 77%, and 85% - 97% for M20 - M50 concrete specimens exposed to a 120-minute duration of heating. The flexural strength was susceptible to higher temperatures.
- After 120 minutes of heating, the residual flexural strength of concrete fell to 0% for all the grades of concrete, and the compressive and tensile strength was higher than that of the flexural strength of concrete.
- The influence of the water-cement ratio played a key role in morphological changes in concrete. While heating duration increased, surface spalling, crack width and weight loss of concrete increased gradually. And the damage evaluation was confirmed with the SEM and image analysis.
- Weight loss was found to increase along with the heating duration and strength grade of concrete. For the concrete specimens exposed to 240 minutes of heating, the weight loss of M20 and M50 concrete was 10.36% and 13.16%, respectively.
- Overall, the higher water-cement ratio concrete performed better than the lower water-cement ratio concrete after exposure to elevated temperature.
- A simple empirical relationship was developed using regression analysis to determine the residual strength by varying the duration of heating and grade of concrete.
(Q7) English Language should be improved throughout the manuscript.
(A7) We thank reviewer 1 for the valuable comment. As per the suggestions, the entire manuscript is checked for the English language and updated in the revised manuscript. It is highlighted in red colour throughout the manuscript.
(Q8) The word "test" from the caption of heading 3.1 should be deleted as this heading talks about results. Similarly, procedures of all tests being performed should be briefly explained in section 2.
(A8) We thank reviewer 1 for the valuable comment. As per the reviewer’s comment, the word “test” from heading 3.1 is changed to “Ultrasonic Pulse Velocity” and updated in the revised manuscript. The procedures of all tests are briefly explained in section 2.
(Q9) The test procedure of results presented in section 3.1 should be explained in section 2.
(A9) We thank reviewer 1 for the valuable comment. The testing procedures of results presented in section 3.1 are now briefly explained in section 2.
Test Procedure:
2.4.1. Ultrasonic pulse velocity test
The UPV test was carried out to analyse the homogeneity and pore density of concrete. Concrete cube specimens were preferred to conduct the UPV. The direct UPV test method was chosen for better results. The experiment was conducted as per IS 516 (2019) [26]. A unipan 543 digital instrument and 40 kHz point ultrasonic heads were used for the test. The plus velocity was estimated based on the specimen dimension and measured transit time of the specimen.
(Q10) Lines 454-455: how much loss is observed and what could be the reason?
(A10) We thank reviewer 1 for the valuable comment. As per the comment by the reviewer, the respective reasoning for Lines 454-455, "higher grade of concrete (M-50) exhibits higher strength loss than M-20 and M-30 grades of concrete" has been given below in detail. The conclusion is rewritten by mentioning the percentage losses observed during the test. However, a detailed discussion is given for reasoning based on past studies and our own observations from the previous research work.
Reason for loss in strength of higher-grade concrete specimens:
The strength loss of heated concrete specimens seems to be directly related to the grade of the concrete. At first 15 minutes, all the specimens showed a reduction in strength; M20 grade specimens exhibited a higher reduction. Concrete specimens with a lower water-cement ratio (M50) had a lower reduction in strength. At 15 minutes, most free water evaporates by leaving free pores in concrete. For concrete specimens with a higher water-cement ratio, heating produces a higher number of pores, thus weakening the internal structure. Beyond 15 minutes of heating and up to 30 minutes, all the specimens show a gain in strength. M50 concrete specimens were found to have the maximum strength gain. Therefore, based on experiments up to 30 min duration of heating, the performance of concrete with a lower water-cement ratio (M50) was found to be good. The lower water-to-cement ratio for M50 concrete makes the concrete less hydrated. Due to this, the availability of a higher amount of unreacted cement particles undergoes rehydration at 30 minutes. The powerful elimination of water (as steam) at high temperatures affected the space enclosed around the cement paste. Due to the resistance of steam to flow, steam generates a high pressure in the paste. Thus, the internal autoclaving process occurred in the cement paste, and as a result, secondary hydration took place on unhydrated cement particles, which caused a small gain in strength [46].
Beyond 30 minutes, M50 concrete specimens were found to have a higher strength reduction than other specimens. The main reason for this steep reduction is due to the evaporation of water from concrete. At this stage, strength loss occurred mainly because of the coarsening of pores. High-strength concrete has fewer pores, and hence dissipation of gasses during heating becomes more difficult. During heating, the vapour pressure exceeds the ultimate tensile strength of concrete and thus leads to the widening of cracks. For lower-grade concrete, pores are interconnected; as a result, these pores act as capillary tubes and help in the easy dissipation of gases. The size of pores increases at 120 minutes duration of heating. With a further increase in the duration of heating from 60 min (925°C) to 120 min (1029°C), the loss of strength became more significant. The duration of exposure above 60 min (925°C) is characterized by the decomposition of Ca(OH)2. Complete dehydration occurs at about 120 min of exposure time to fire, and dehydration of C-S-H gel occurs in cement paste, and the paste loses its cementing ability. At this duration, due to high temperature, the crystal structure transformation of aggregate takes place, and thus, aggregate losses its strength.
At a heating duration above 180 min (1090°C), the concrete specimens lose about 85–90% of the initial strength at 240 minutes of heating. At 240 minutes, the strength reduction rate is almost the same for all grades of concrete. Between 180 min and 240 min duration of heating, the effect of heating duration and temperature surpasses the effect of the grade of concrete on the strength loss of concrete.
The effect of the grade of concrete on strength loss is also confirmed by spalling observation. The extent, severity, and nature of spalling vary as exposure duration increases. Based on the experimental results, it is found that the significant factors contributing to spalling are density and the moisture content present in the concrete. Generally, concrete with the higher w/c ratio (M20-M30) has greater porosity; for this reason, it does not show any explosive spalling behaviour but surface pitting. Spalling was considered insignificant when only surface pitting occurred. However, spalling can affect a structural element's fire resistance and load-bearing capacity if it exposes the core and reinforcing steel due to a rapid rise in temperature. Thus, the consequences of spalling depend upon the application for which the concrete is used. On the contrary, progressive and explosive spalling takes place for concrete with a lower w/c ratio (M40-M50) due to its higher dense structure. The explosive spalling occurred between 60 and 120 minutes of heating (This is evident based on the explosive sound heard during the experiment).
Surface pitting was observed in the M20 (60 min-120 min), M30 (60 min-120 min), and M40 (60 min) grade concrete specimens. The reason for the surface pitting is due to the thermal cracking at the concrete surface. The shrinkage of cement paste causes thermal cracking due to the evaporation of water from the surface, a thermal mismatch between aggregate and cement paste, and thermal gradients between the interior and exterior layer of concrete (Fu and Li, 2011; Al Qadi and Al-Zaidyeen, 2014).
Significant spalling behaviour is noticed in the M40 and M50 grade concrete specimens. Progressive spalling (surface spalling and corner spalling) was observed in the M40 (120 min) and M50 (60 min) grade concrete specimens. The thermal shock from severe thermal gradients induces high compressive stresses close to the heated surface due to restrained thermal expansion and tensile stresses in the cooler interior regions (Bazant, 1997). Local strain incompatibilities between the cement paste and the aggregates also exist simultaneously. Further, explosive spalling was observed in the M50 grade concrete specimens at 120 min duration of heating. This is due to the dense microstructure in the higher-grade concrete specimens. The reason for the explosive spalling is the very rapid rise of the temperature in the furnace, which creates a large thermal gradient between the surface and the internal core of the concrete specimen due to the low conductivity and high heat capacity of concrete. With increasing temperatures, due to steep thermal gradients, some water evaporates through the surface, and the remaining is transferred into the interior region. During this process, pore pressures are generated in the pore network. The increase in temperature causes a rapid rise in the pore pressure in the concrete. This high pore pressure creates tensile stress in the concrete. Explosive Spalling occurs when the tensile stress in concrete caused by this pore pressure exceeds the tensile strength of the concrete. It was reported by Hertz (2003) that the high strength concrete is denser, and hence considerable internal pressure gets built up, leading to spalling of concrete.
Generally, spalling occurs due to the development of pore pressure in dense concrete. Even though cube and cylinder specimens have the same moisture content, density, and porosity, the critical temperature required for developing pore pressure first reaches the cylinder specimen due to symmetrical heat distribution. However, in the case of cube specimens, the vapour pressure was released through thermal cracks [Fu and Li, 2011]. This may be the reason for cylinder specimens having more spalling than cube specimens, as per the observation based on the present study.
(Q11) Lines 456-457: how much increase is there and what could be the reason? The word "significantly" seems a little vague in conclusion. Similarly, remaining conclusions need major rephrasing with scientific soundness.
(A11) We thank reviewer 1 for the valuable comment. As per the reviewer’s opinion, the reason for lines 456-457, and the word "the strength loss of concrete increases significantly as the heating intensity increases" has been given below in detail. The conclusions have been rephrased and revised in the manuscript. However, a detailed discussion is given for reasoning based on past studies and our own observations from the previous research work.
Reason for the strength loss of concrete increases significantly as the heating intensity increases:
As the strength grade of concrete increases, the compressive strength of the reference specimen increases. The strength of heated specimens was found to be in decreasing order compared to reference specimens as heating durations increased. This shows that as duration and temperature increase, dehydration and decomposition of the cementitious compound increase from the surface to the interior layer of the specimen. The high temperature of concrete structures mainly reduces the compressive strength of concrete (Savva et al., 2005; Poon et al., 2004; Xiao and Konig, 2004; Li et al., 2003). This drop in strength is due to dehydration and decomposition of internal chemical structure in cement paste as the exposure duration and temperature increase (Chan and Peng, 1999; Lin et al., 1996). The target temperature up to 60 min is 925°C. At this stage, it was observed from the investigation that a partial change in colour is observed, and also cracks on the specimen’s surface are found to be lower.
For the specimens heated up to 120 min, the residual strength is lower than those heated up to 60 min. This is due to the higher duration of heating and higher temperature to which specimens are exposed. Due to this, concrete with different strength grades was damaged severely. However, the reduction in strength is higher for concrete with M50 grade. It can also be observed during the investigation that severe spalling occurred in this stage.
The reduction in strength is more pronounced based on moisture content, density, heating rate, and porosity. The weakening of cement paste causes the disintegration of concrete. Hydrated cement paste contains CSH, CH, and ettringite. At 100°C, normally free water and absorbed water of the cement paste gets evaporated (Savva et al., 2005). A desiccation occurs in this state, and it is the reason for cracks in the surface. But above 400°C, chemically combined water also started losing at this state. The literature has reported that the strength loss is largely attributed to the decomposition of calcium hydroxide, which generally occurs over 500°C (Khaliq and Kodur, 2011).
The specimens heated for 180 (1090°C) and 240 min (1133°C) duration show almost the same damage level. The difference in the residual strength for all the grades of concrete is marginal. All the concrete specimens subjected to 1090°C and 1133°C suffered extreme damage. The heated concrete became so brittle. The density of thermal cracks was also higher for the specimens subjected to 120 min duration of heating than specimens subjected to 60 min and 120 min duration of heating. Also, a complete colour change is observed in this stage.
The loss of water mainly depends on the heating duration and temperature magnitude. Due to this loss, strength also started deteriorating in the concrete. Normally above 500°C, dehydration, and disintegration of CSH gel occur. Also, remarkable chemical changes take place in the cement paste. Generally, over 900°C, a complete decomposition takes place in the concrete, and thereby concrete losses its strength, as reported by Ilker Bekir Topcu et al. (2011). The reduction in strength is due to the dehydration of absorbed water and chemically bound water (Arioz, 2007). The decrease in strength may be due to decomposition, deterioration, and thermal incompatibility, as reported by Xiong and Liew (2015).

Reviewer 2 Report
· The informal language is not suitable and should be improved extensively. The article needs major grammatical and syntax improvements. Use of English service center is recommended. Several sentences are not clear and understandable.
· Majority of the qualitative statements should be modified for quantified result comparisons.
· The introduction needs to be revised for higher quality language. The author mentioned some works without stating about the contributions, pros and cons and the how the current work would address.
· The purpose of the article should be clarified in details, why and where this study could be beneficent, more in depth conclusion should be provided.
· The authors mentioned “When the concrete is subjected to temperature exposure, the hardening performance of the concrete tends to degrade drastically ” The following references should be added for comprehensiveness of this statement 1) Compressive behavior of concrete under environmental effects. IntechOpen. 2) Temperature and humidity effects on behavior of grouts. Advances in concrete construction
· The authors mentioned “Normal strength concrete (NSC) is used primarily in many earlier studies as a research area” The following references should be added for comprehensiveness of this statement 1) Experimental investigation of sound transmission loss in concrete containing recycled rubber crumbs. 2) Nano silica and metakaolin effects on the behavior of concrete containing rubber crumbs. CivilEng. 3) Investigation of steel fiber effects on concrete abrasion resistance, Advances in concrete construction.
· The selection of the specimen and properties should be justified.
· More in depth conclusions should be drawn based on various studies, the summary should indicate in depth results and conclusions.
· Any figures taken from other works should be reestablished and referenced.
· All Abbreviations should be expanded.
· How the R2 values for M20 regression analysis is relatively low.
· Why the cement to water ratio of 0.58 is chosen which is significantly high affecting the mechanical properties of concrete.
· From Fig 9., how the high duration has highest weight loss for M50 while for low duration M20 has the highest weight loss.
·
Author Response
Responses to Reviewer comments
The authors sincerely thank the editor and reviewer 2 for the constructive comments. All of the comments have been carefully considered and, wherever appropriate, revisions have been made to the manuscript. Responses to these comments and revisions implemented in the paper are detailed below. The Reviewer comments are in italic black fonts while our replies are in blue. The text changed in the manuscript is highlighted in red.
Reviewer # 2
Comments and Suggestions for Authors
(Q1) The informal language is not suitable and should be improved extensively. The article needs major grammatical and syntax improvements. Use of English service center is recommended. Several sentences are not clear and understandable.
(A1) We thank reviewer 2 for the valuable comment. As per the reviewer’s suggestion, the entire manuscript is checked for grammatical/spelling errors, and a number of sentences sentences are revised.
(Q2) Majority of the qualitative statements should be modified for quantified result comparisons.
(A2) We thank reviewer 2 for the valuable comment. As per the reviewer’s suggestion, modifications have been made and presented in the manuscript. The given discussions are updated in the appropriate section of the manuscript in red.
Beyond 30 minutes, a higher reduction occurred in M-50 concrete specimens than in other specimens. This drop in strength is due to dehydration and decomposition of internal chemical structure in cement paste as the exposure duration and temperature increase (Chan et al., 1999; Lin et al., 1996). The reduction in strength is more pronounced based on moisture content, density, heating rate, and porosity. The weakening of cement paste causes the disintegration of concrete. Hydrated cement paste contains CSH, CH, and ettringite. At 100°C, normally free water and absorbed water of the cement paste gets evaporated (Savva et al., 2005). A desiccation occurs in this state, and it is the reason for cracks in the surface. But above 400°C, chemically combined water also started losing at this state. The literature has reported that the strength loss is largely attributed to the decomposition of calcium hydroxide, which generally occurs over 500°C (Khaliq and Kodur, 2011).
From 180 to 240 minutes of exposure, the effect of heating duration and temperature surpasses the effect of CS of concrete on the strength loss of specimens. Normally above 500°C, dehydration and disintegration of CSH gel occur. Also, remarkable chemical changes take place in the cement paste. Generally, over 900°C, a complete decomposition takes place in the concrete, and thereby concrete losses its strength, as reported by Topcu et al. (2011). The reduction in strength is due to the dehydration of absorbed water and chemically bound water (Arioz, 2007). The decrease in strength may be due to decomposition, deterioration, and thermal incompatibility, as reported by Xiong and Liew (2015).
The reduction in TS was higher for specimens with higher grades than lower grades at all heating durations, and an additional strength loss was noticed. It has been speculated (Fares et al., 2009; Kalifa et al., 2000; Phan et al., 2001; Bastami et al., 2011; Sideris et al., 2009) that the higher density in high-strength concrete restricts the water vapour pressure from escaping. This internal pore pressure exceeds the concrete's tensile strength, resulting in large numbers of micro and macro cracks (Sideris, 2007; Chan et al., 1999; Khoury, 1992).
So, crack formation under flexure was found to be more critical than compression. An increase in temperature exposure increases crack propagation; this might be attributed to the degradation of CSH gel phases (Correia et al.,2014). Heated specimens are more prone to crack under flexural loading and close up under compressive loading. Thus, the impact of crack coalescence is more crucial on the flexural strength than that of the compressive strength of concrete specimens (Potha Raju et al., 2004; Aydın and Baradan, 2007).
References:
Chan, Y.N., Peng, G.F. and Anson, M. (1999), “Residual strength and pore structure of high-strength concrete and normal strength concrete after exposure to high temperatures”, Cement and Concrete Composites, Vol. 21 No. 1, pp. 23-27.
Lin, W.M., Lin, T.D. and Powers-Couche, L.J. (1996), “Microstructures of fire-damaged concrete”, Materials Journal, Vol. 93 No. 3, pp. 199-205.
Savva, A., Manita, P. and Sideris, K.K. (2005), “Influence of elevated temperatures on the mechanical properties of blended cement concretes prepared with limestone and siliceous aggregates”, Cement and Concrete Composites, Vol. 27 No. 2, pp. 239-248.
Khaliq, W. and Kodur, V. (2011), “Thermal and mechanical properties of fiber reinforced high performance self-consolidating concrete at elevated temperatures”, Cement and Concrete Research, Vol. 41 No. 11, pp. 1112-1122.
Topcu, I.B., Boga, AR and Demir, A. (2011), “Influence of cover thickness on the mechanical properties of steel bar in mortar exposed to high temperatures”, Fire and Materials, Vol. 35 No. 2, pp. 93-103.
Arioz, O. (2007), “Effects of elevated temperatures on properties of concrete”, Fire safety journal, Vol. 42 No. 8, pp. 516-522.
Xiong, M.X. and Liew, J.R. (2015), “Spalling behavior and residual resistance of fibre reinforced Ultra-High performance concrete after exposure to high temperatures”, Materiales de Construccion, Vol. 65 No. 320, pp. 071.
Fares, H., Noumowe, A. and Remond, S. (2009), “Self-consolidating concrete subjected to high temperature: mechanical and physicochemical properties”, Cement and Concrete Research, Vol. 39 No. 12, pp. 1230-1238.
Kalifa, P., Menneteau, F.D. and Quenard, D. (2000), “Spalling and pore pressure in HPC at high temperatures”, Cement and concrete research, Vol. 30 No. 12, pp. 1915-1927.
Phan, L.T., Lawson, J.R. and Davis, F.L. (2001), “Effects of elevated temperature exposure on heating characteristics, spalling, and residual properties of high performance concrete”, Materials and Structures, Vol. 34 No. 2, pp. 83-91.
Bastami, M., Chaboki-Khiabani, A., Baghbadrani, M. and Kordi, M. (2011), “Performance of high strength concretes at elevated temperatures”, Scientia Iranica, vol. 18 No. 5, pp. 1028-1036.
Sideris, K.K., Manita, P. and Chaniotakis, E. (2009), “Performance of thermally damaged fibre reinforced concretes”, Construction and Building Materials, Vol. 23 No. 3, pp. 1232-1239.
Sideris, K.K. (2007), “Mechanical characteristics of self-consolidating concretes exposed to elevated temperatures”, Journal of materials in civil engineering, Vol. 19 No. 8, pp. 648-654.
Khoury, G.A. (1992), “Compressive strength of concrete at high temperatures: a reassessment”, Magazine of concrete Research, Vol. 44 No. 161, pp. 291-309.
Correia JR, Lima JS and de Brito J. Post-fire mechanical performance of concrete made with selected plastic waste aggregates. Cement and Concrete Composites 2014; 53: 187-199.
Potha Raju M, Shobha M and Rambabu K. Flexural strength of fly ash concrete under elevated temperatures. Magazine of Concrete Research 2004; 56: 83-88.
Aydın S and Baradan B. Effect of pumice and fly ash incorporation on high temperature resistance of cement based mortars. Cement and Concrete Research 2007; 37: 988-995.
(Q3) The introduction needs to be revised for higher quality language. The author mentioned some works without stating about the contributions, pros and cons, and the how the current work would address.
(A3) We thank reviewer 2 for the valuable comment. As per the comment, the introduction section is carefully revised with the present research scope, checked English quality, and updated in the revised manuscript.
Introduction:
Fire has always been an incessant threat to the stability of a structure. It represents a vulnerable factor of great hazard that can create havoc to buildings and civil infrastructures. Countless fire accidents occur around the world and they lead to the degradation of the essential qualities of a sound infrastructure. The load-carrying capacity of the building decreases significantly and leads to the collapse of the structure due to the degradation of the strength properties of the building materials [2].
The compressive strength of concrete is one of the primary factors used in the design of reinforced concrete structures. At the given temperature conditions, the compressive strength of concrete is based on the w/c ratio, cementitious material, aggregate type, and curing conditions [5]. At a higher temperature, a higher reduction of tensile strength and flexural strength of concrete was observed compared to its compressive strength [6]. The decline in the strength of the concrete is due to the decomposition of CSH gel, degradation of calcium hydroxide (Ca (OH)2), decomposition of cement paste, and deformation of aggregates [7].
Studies related to behavioural changes of different grades of concrete mixes under elevated temperature exposure are essential to understand their levels of degradation to enhance their performance during fire accidents. When the concrete gets exposed to elevated temperature, its internal microstructure gets distorted and results in a drastic reduction in the strength and durability of the concrete. The structural and thermal behaviour of the varied grades of concrete subjected to different temperature loads will give an overview of how they respond to elevated temperatures. And hence it becomes a prerequisite to study and understand the behaviour of structural materials subjected to fire and their mechanical response to heating.
The extent of damage arising in the fire-affected concrete depends on the magnitude of temperature, concrete mix and the cooling type adopted for bringing down the fire. During fire accidents, generally, concrete structures are quenched by water extinguishers. Thus, estimating the damage level of water-cooled concrete structures is vital to evaluate the residual strength capacity for repair and rehabilitation works. The study aims to set a database for the residual hardening performance of concrete subjected to standard fire condition and to anticipate the expected changes in the mechanical behaviour of concrete exposed to very high temperatures followed by fast cooling procedures.
There is a scarcity of fundamental knowledge on the influence of standard fire on the thermal characteristics of concrete, notably for concrete with varying strength grades. While knowledge and experience with various types of concrete behaviour at ambient and increased temperatures are well established, the effect of water cooling on concrete with varied grades of concrete subjected to typical fire requires further investigation. Since the residual hardening performance for water-cooled specimens differs significantly from air-cooled specimens, this data will be helpful in formulating structural fire standards for better resistant properties. The data on the physical, mechanical, and microstructure properties of water-cooled specimens may be relevant in predicting the evacuation time of humans before the structural collapse during building fires.
(Q4) The purpose of the article should be clarified in details, why and where this study could be beneficent, more in depth conclusion should be provided.
(A4) We thank reviewer 2 for the valuable comment. A separate brief section explaining the purpose of this research for practical implementation has been added and updated in the manuscript. Also, conclusions have been made short and precise with valid mentioning of results and updated in the revised manuscript.
Relevance of the proposed research for practical implementation:
Fire poses a significant threat to life safety even under a perfect shelter for humanity. Every minute is critical in evacuating inhabitants when a fire breaks out in a building. Conducting experiments on the fire resistance capacity of materials is too risky and costly. Therefore, the residual strength data on concrete will be helpful in developing different applications for safety and security. Experimental data extracted from investigations on strength capacity may be useful for the analytical modelling of modified structural members under elevated temperatures as an input to the finite element model. Material strength values are also essential for estimating the strength of reinforced members subjected to various heating durations. This enables in adopting of a suitable repair technique to improve the durability of fire-affected buildings. It will be useful to identify the appropriate grade of concrete for ample fire endurance, as it is an important design parameter. Further, developing an empirical relationship between residual strengths will become beneficial for design applications. The data on residual strength gives an idea about the reduction in material safety during fire exposure, which will provide the guideline for preparing the preliminary design for civil/structural engineers.
(Q5) The authors mentioned “When the concrete is subjected to temperature exposure, the hardening performance of the concrete tends to degrade drastically” The following references should be added for comprehensiveness of this statement 1) Compressive behavior of concrete under environmental effects. IntechOpen. 2) Temperature and humidity effects on behaviour of grouts. Advances in concrete construction
(A5) We thank reviewer 2 for the valuable comment. As per the comment, the mentioned literature is cited and updated in the revised manuscript.
When the concrete is subjected to temperature exposure, the hardening performance of the concrete tends to degrade drastically. So, the deterioration in the quality of concrete occurs [3,4].
[3]. Farzampour, A. Compressive Behavior of Concrete under Environmental Effects. Compressive strength of concrete, 92-104. DOI: 10.5772/intechopen.85675.
[4]. Farzampour, A. Temperature and humidity effects on behavior of grouts. Advances in concrete construction 2017, 5(6), 659. doi.org/10.12989/acc.2017.5.6.659.
(Q6) The authors mentioned “Normal strength concrete (NSC) is used primarily in many earlier studies as a research area” The following references should be added for the comprehensiveness of this statement 1) Experimental investigation of sound transmission loss in concrete containing recycled rubber crumbs. 2) Nano silica and metakaolin effects on the behavior of concrete containing rubber crumbs. CivilEng. 3) Investigation of steel fiber effects on concrete abrasion resistance, Advances in concrete construction.
(A6) We thank reviewer 2 for the valuable comment. As per the comment, the mentioned literature is cited and updated in the revised manuscript.
Normal strength concrete (NSC) is used primarily in many earlier studies as a research area [15-18].
[15]. Chalangaran, N.; Farzampour, A.; Paslar, N.; Fatemi, H. Experimental Investigation of Sound Transmission Loss in Concrete Containing Recycled Rubber Crumbs.
[16]. Chalangaran, N.; Farzampour, A.; Paslar, N. Nano Silica and Metakaolin Effects on the Behavior of Concrete Containing Rubber Crumbs. Civil Eng 2020, 1, 264-274. https://doi.org/10.3390/civileng1030017.
[17]. Iman Mansouri.; Farzaneh Sadat Shahheidari.; Seyyed Mohammad Ali Hashemi.; Alireza Farzampour. Investigation of Steel Fiber Effects on Concrete Abrasion Resistance. Adva in Con Cons 2020,9,367-374. doi.org/10.12989/acc.2020.9.4.367.
[18] Kiran, T.; Anand, N.; Mathews, M.E.; Kanagaraj, B.; Andrushia, A.D.; Lubloy, E.; G, J. Investigation on Improving the Residual Mechanical Properties of Reinforcement Steel and Bond Strength of Concrete Exposed to Elevated Temperature. Case Stud. Constr. Mater. 2022, 16, e01128, doi:10.1016/j.cscm.2022.e01128.
(Q7) The selection of the specimen and properties should be justified.
(A7) We thank reviewer 2 for the valuable comment. As per the comment, the specimen and properties are explained and updated in the revised manuscript.
After the cooling treatment, experiments are conducted to find the concrete's stress-strain behavior, FS, CS, and TS. An average of three samples was considered in the present investigation to achieve accuracy. A detailed experimental procedure is elaborated below. The test images are illustrated in Figures 2a, 2b, and 2c.
2.4.1. Ultrasonic pulse velocity test
The UPV test was carried out to analyse the homogeneity and pore density of concrete. Concrete cube specimens were preferred to conduct the UPV. The direct UPV test method was chosen for better results. The experiment was conducted as per IS 516 (2019) [26]. A unipan 543 digital instrument and 40 kHz point ultrasonic heads were used for the test. The plus velocity was estimated based on the specimen dimension and measured transit time of the specimen.
2.4.2. Stress-Strain behaviour of concrete test
The stress strain behaviour of concrete was evaluated, with the cylinder specimen dimension of 150 mm diameter and 300 mm length. The experiment was performed by using universal testing machine. The equipment is well designed such way that to record the failure load of concrete and stress strain behaviour of test specimens. The testing program was done according to IS 516:2004 [26]. Figure 2(a) shows the testing of cylinder samples.
2.4.3. Tensile strength test
The experiment was conducted to evaluate the tensile strength of concrete as per the guidelines of IS 5816:2004 [27]. The cylinder concrete specimen having 150mm diameter and 300 mm length was used for testing. The tests were performed under CTM equipment, the load was applied at a rate of 2 N/mm2/min. The test readings were noted till the specimen failure. Figure 2(b) shows the testing of samples for obtaining tensile strength.
2.4.4. Flexural strength test
Detailed experiment is conducted to evaluate the flexural strength of concrete. Prism specimen of width 100mm, depth 100 mm and length 500 mm were used for the testing. The test standards were followed as per IS 516:2004 [26]. Samples are tested under the two-point loading setup. A constant load was applied at constant rate of 180 kg/min till failure of specimen. Figure 2(c) shows the flexural strength test of concrete prism.
Fig. 2 View of (a) CS test, (c) TS, and (d) FS test of concrete
Reference:
- IS 516 (1999), Indian Standard Code of Practice for Method of Tests for Strength of Concrete, (Reaffirmed 2004). Bureau of Indian Standards, New Delhi.
- IS 5816 (1999), Indian Standard Code of Practice for Method of Tests for Splitting Tensile Strength of Concrete, (Reaffirmed 2004). Bureau of Indian Standards, New Delhi.
(Q8) More in depth conclusions should be drawn based on various studies, the summary should indicate in depth results and conclusions.
(A8) We thank reviewer 2 for the valuable comment. Accordingly, conclusions have been made short and precise with valid mentioning of results and updated in the revised manuscript.
A detailed investigation was carried out to evaluate the residual strength properties of different grades of concrete mixes after being cooled by water. The strength properties such as CS, TS, and FS were examined. The physical characteristics and morphological changes were analysed. Based on the experimental investigations conducted on the grade of concrete and the effect of elevated temperature, the following conclusions were drawn:
- As the duration of heating increased, the UPV values of heated concrete specimens declined for all the grades of concrete. The UPV result showed poor quality in concrete after 60-minute exposure and zero values at 180 and 240-minute exposure. At 45 minutes, the UPV values of M-20 and M30 specimens showed doubtful quality, whereas, M-40 and M50 specimens exhibited good quality.
- After exposure to a prolonged duration of beyond 60 minutes, concrete specimens with higher grade (M50) exhibited a higher rate of strength loss or weight loss compared to lower strength grade (M20).
- The loss in compressive strength, tensile strength, and flexural strength ranged from 22% - 30%, 54% - 69%, and 70% - 88% for M20 - M50 concrete specimens exposed to a 60-minute duration of heating.
- The loss in compressive strength, tensile strength, and flexural strength ranged from 51% - 65%, 70% - 77%, and 85% - 97% for M20 - M50 concrete specimens exposed to a 120-minute duration of heating. The flexural strength was susceptible to higher temperatures.
- After 120 minutes of heating, the residual flexural strength of concrete fell to 0% for all the grades of concrete, and the compressive and tensile strength was higher than that of the flexural strength of concrete.
- The influence of the water-cement ratio played a key role in morphological changes in concrete. While heating duration increased, surface spalling, crack width and weight loss of concrete increased gradually. And the damage evaluation was confirmed with the SEM and image analysis.
- Weight loss was found to increase along with the heating duration and strength grade of concrete. For the concrete specimens exposed to 240 minutes of heating, the weight loss of M20 and M50 concrete was 10.36% and 13.16%, respectively.
- Overall, the higher water-cement ratio concrete performed better than the lower water-cement ratio concrete after exposure to elevated temperature.
- A simple empirical relationship was developed using regression analysis to determine the residual strength by varying the duration of heating and grade of concrete.
(Q9) Any figures taken from other works should be reestablished and referenced.
(A9) We thank reviewer 2 for the valuable comment. All the figures are taken from the original work of our research, a study conducted in our university's structural fire engineering lab. Some references from our previous studies is listed below which is also cited in the manuscript:
- Kiran, T.; Yadav, S.K.; N, A.; Mathews, M.E.; Andrushia, D.; lubloy, E.; Kodur, V. Performance Evaluation of Lightweight Insulating Plaster for Enhancing the Fire Endurance of High Strength Structural Concrete. Build. Eng. 2022, 57, 104902, doi:10.1016/j.jobe.2022.104902.
- Kiran, T.; Anand, N.; Mathews, M.E.; Kanagaraj, B.; Andrushia, A.D.; Lubloy, E.; G, J. Investigation on Improving the Residual Mechanical Properties of Reinforcement Steel and Bond Strength of Concrete Exposed to Elevated Temperature. Case Stud. Constr. Mater.2022, 16, e01128, doi:10.1016/j.cscm.2022.e01128.
- Anand, N.; Andrushia, A.D.; Kanagaraj, B.; Kiran, T.; Chandramohan, D.L.; Ebinezer, S.; Kiran, R.G. Effect of Fibers on Stress–Strain Behavior of Concrete Exposed to Elevated Temperature. Today Proc. 2022, 60, 299–305, doi:10.1016/j.matpr.2022.01.223.
- Kiran, T.; Anand, N.; Mathews, M.E.; Kanagaraj, B.; Andrushia, A.D.; Lubloy, E.; G, J. Investigation on Improving the Residual Mechanical Properties of Reinforcement Steel and Bond Strength of Concrete Exposed to Elevated Temperature. Case Stud. Constr. Mater. 2022, 16, e01128, doi:10.1016/j.cscm.2022.e01128.
- Kanagaraj, B.; Anand, N.; Andrushia, A.D.; Lubloy, E. Investigation on Engineering Properties and Micro-Structure Characteristics of Low Strength and High Strength Geopolymer Composites Subjected to Standard Temperature Exposure. Case Stud. Constr. Mater. 2022, 17, e01608, doi:10.1016/j.cscm.2022.e01608.
- Kanagaraj, B.; Anand, N.; Johnson Alengaram, U.; Samuvel Raj, R.; Kiran, T. Exemplification of Sustainable Sodium Silicate Waste Sediments as Coarse Aggregates in the Performance Evaluation of Geopolymer Concrete. Build. Mater. 2022, 330, 127135, doi:10.1016/j.conbuildmat.2022.127135.
- Thanaraj, D.P.; Anand, N.; Arulraj, G.P. Post-Fire Damage Assessment and Capacity Based Modeling of Concrete Exposed to Elevated Temperature; 2019; Vol. 0; ISBN 1056789519881.
- Thanaraj, D.P.; Anand, N.; Arulraj, P. Strength and Microstructure Characteristics of Concrete with Different Grade Exposed to Standard Fire. Struct. Fire Eng. 2020, 11, 261–287, doi:10.1108/JSFE-09-2018-0021.
(Q11) All Abbreviations should be expanded.
(A11) We thank reviewer 2 for the valuable comment. We have added the abbreviation list in a table form in the revised manuscript.
List of abbreviations and symbols
|
CS |
Compressive Strength |
|
TS |
Tensile Strength |
|
FS |
Flexural Strength |
|
CA |
Coarse Aggregate |
|
WL |
Weight loss |
|
CSH |
Calcium Silicate Hydrate |
|
Ca(OH)2 |
Calcium hydroxide |
|
w/c ratio |
Water to cement ratio |
|
M-20 |
Concrete with strength 20 MPa |
|
M-30 |
Concrete with strength 30 MPa |
|
M-40 |
Concrete with strength 40 MPa |
|
M-50 |
Concrete with strength 50 MPa |
|
NSC |
Normal strength concrete |
|
HSC |
High strength concrete |
|
fck |
Characteristic compressive strength of concrete (MPa) |
|
fck(t) |
Residual compressive strength |
|
fct(t) |
Residual tensile strength |
|
fcr(t) |
Residual flexural strength |
|
C-H |
Calcium hydroxide |
(Q12) How the R2 values for M20 regression analysis is relatively low.
(A12) We thank reviewer 2 for the valuable comment. The reduction in compressive strength of M20 strength grade concrete is considerably lower, i.e., 24%, at a 15-minute duration of heating. The reduction in compressive strength at 15-minute exposure is higher than the reduction at 60-minute heating exposure, which is 22%. During first 15 minutes, all the specimens showed a reduction in strength; M20 grade specimens exhibited a higher reduction. Concrete specimens with a lower water-cement ratio (M50) had a lower reduction in strength. At 15 minutes, most of the free water evaporates by leaving free pores in the concrete. For concrete specimens with a higher water-cement ratio, heating produces a higher number of pores, thus weakening the internal structure.
A sudden fall in the curve is noticed in the residual compressive strength figure, which makes the correlation point deviate to a larger extent from the scattered regression analysis curve. Due to this, the R2 values are relatively low for lower grade concrete compared to higher grade concrete which is seen in Figure 1, Correlation between compressive strength and UPV values.
Fig. 1. Correlation between compressive strength and UPV of M-20, M-30, M-40 and M-50 concrete
(Q13) Why the cement to water ratio of 0.58 is chosen which is significantly high affecting the mechanical properties of concrete.
(A13) We thank reviewer 2 for the valuable comment. As the authors have used the manufactured sand (M-Sand) to overcome the water absorption of M-Sand and to attain the target slump, this ratio was selected. However, this was helpful for the authors to compare the effect of elevated temperature on the low-strength concrete having higher porosity and concrete with higher strength grade having lesser porosity.
(Q14) From Fig 9., how the high duration has highest weight loss for M50 while for low duration M20 has the highest weight loss.
(A14) We thank reviewer 2 for the valuable comment. In accordance with the comment, the explanation for Figure 9 in the manuscript is given below. A detailed discussion is given for reasoning based on past studies and our own observation from the previous research work. The relevant discussions are updated in the appropriate section in the manuscript.
The scattered data in Figure 9 (in manuscript) has two distinct gradient patterns, the initial sharp gradient from 0 min to 60 min exposure time followed by a flatter gradient above 60 min duration of heating. The initial weight loss was due to the evaporation of absorbed and chemically bound water, which is also confirmed by TGA analysis. The weight loss on the specimens heated up to 15 min (718°C) duration is negligible because the evaporation of water just begins at this duration of heating. At this duration, the M20 concrete specimen experienced a weight loss of 2.44%, and the M50 concrete specimen experienced a weight loss of 1.24%. The effect of the w/c ratio on the weight loss of heated concrete specimens is significant. The weight loss of M20 concrete is found to be higher, i.e., about 5.47% up to 60min. This is due to the higher water content in the concrete, which creates more pores during heating. However, for M50 concrete, it is 4.55%.
Beyond 60 min (925°C) fire exposure time, weight loss gradually increased because of the decomposition of Ca(OH)2, C–S–H, and CH hydrates. At 240 min (1133°C) duration of heating, the losses were about 9.56% for M20 concrete and 13.16% for M50 concrete. Beyond 60 min duration of heating, it is observed that the weight loss is found to be higher for M50 concrete. This is due to the lower water-cement ratio resulting in a less porous structure and higher cement content. The higher amount of cement content leads to a larger decomposition of Ca(OH)2, C–S–H, and CH hydrates.
The changes in the composition of the structure can be evaluated by using thermal analysis such as Thermo Gravimetric (TG). The results are also confirmed by Thermo Gravimetric analysis (TG). These data may be useful for understanding the behavior of concrete subjected to high temperatures. Also, for understanding the reasons for strength reduction or weight loss in different heating ranges.
Table 1 gives the details of mass loss for unheated and heated samples
Table 1. Details of Mass Loss
|
Sample |
Temperature range |
Mass loss (%) |
Total loss (%) |
Reason |
|
M20-R |
up to 100°C |
1.52 |
|
Evaporation of free/surface water and dehydration of CSH up to 450°C, Evaporation of strongly bound water and decomposition of Ca(OH)2 up to 700°C, decomposition of C-S-H and Ca(CO)3 occurs beyond 700°C |
|
100°C to 200°C |
1.49 |
|
||
|
200°C to 450°C |
2.10 |
6.44 |
||
|
450°C to 700°C |
0.96 |
|
||
|
above 700°C
|
0.37 |
|
||
|
M50-R |
up to 100°C |
1.25 |
|
|
|
100°C to 200°C |
2.09 |
7.66 |
||
|
200°C to 450°C |
2.15 |
|
||
|
450°C to 700°C |
1.51 |
|
||
|
above 700°C |
0.66 |
|
||
|
200°C to 450°C |
Nil |
Nil |
||
|
450°C to 700°C |
|
|
||
|
above 700°C |
|
|
For the unheated concrete specimen (M20-Reference) the mass loss of about 5.11% at 450°C corresponding to the water which is weakly linked to the hydrates, but in the case of the M50 concrete Reference specimen the mass loss observed was about 5.49% at 450°C. The drop in this range was sharp. Up to 700°C, the drop sustained with a moderate. The mass loss beyond 450°C, maybe because of the discharge of chemically combined water from CSH and Ca(OH)2. The mass loss was found to be 0.96% and 1.51% for M20 Reference and M50 Reference samples, respectively. Hence, the development of cracks and pore pressure and separation in molecular bonds arises in this stage.
Based on the experimental results, it is understood that the significant factors contributing to spalling are density and the moisture content present in the concrete. Spalling of concrete is also a primary reason for higher strength and weight loss for M50 grade concrete at higher temperatures. Table 2 presents the spalling of concrete specimens with a high w/c ratio (M20) and concrete with a low w/c ratio (M50) after exposure to standard fire.
Generally, concrete with a higher w/c ratio (M20-M30) has greater porosity; therefore, it does not show any explosive spalling behaviour. On the contrary, explosive spalling takes place for concrete with a lower w/c ratio (M40-M50) due to its higher dense structure. The explosive spalling occurred between 60 and 120 minutes of heating.
Phan and Carino reported that the tendency for explosive spalling depends on the w/c ratio. The increase in the internal pore pressure owed to the vaporization of the free and chemically bound water cause cracks in concrete and explosive spalling. High-strength concrete is denser, and hence internal pressure gets built up, leading to spalling of concrete. It can be seen from the table that the local spalling occurred for lower-strength grade concrete specimens. But higher strength grade concrete specimens exhibited a sloughing off.
Spalling in concrete specimens may be due to dense concrete structure, sudden temperature increase, and irregular temperature distribution. Exposure to higher temperatures results in a sharp thermal gradient due to concrete's low conductivity and high heat capacity. As a result of this, tensile stresses occur in concrete. These tensile stresses and pore pressure in one direction cause spalling. During heating, thermal compressive stress is also developed at the hot outer layer of the concrete specimen in addition to tensile stresses. If this tensile stress reaches the tensile strength of concrete, the corner piece can spall. It is observed that spalling occurred significantly in cube and cylinder specimens during the heating.
Weight loss and intensity of cracks are directly proportional to the rate of spalling. As the crack-induced spalling increases, the weight loss and density of cracks increase for all the grades of concrete (M20-M50). The weight loss of higher-grade concrete is higher, and the same is confirmed from the experiments. This higher weight loss is observed beyond 120 min (1029°C) duration of heating, and explosive spalling also occurred in the same duration. Larger crack widths and more cracks were observed for concrete with a lower w/c ratio, and these specimens exhibited explosive spalling.
Table 2. Spalling of Concrete Specimens Exposed to Elevated Temperature.
|
M20 (Cube specimen) |
M50 (Cube specimen) |
M20 (Cylinder specimen) |
M50 (Cylinder specimen) |
|
|
|
|
|
Reference:
Phan, L. T., & Carino, N. J. (2002). Effects of test conditions and mixture proportions on behavior of high-strength concrete exposed to high temperatures. ACI Materials Journal, 99(1), 54-66.

Round 2
Reviewer 2 Report
Grammatical aspects could be improved
Author Response
Responses to Reviewer comments
The authors sincerely thank the editor and reviewer 2 for the constructive comments. All of the comments have been carefully considered and, wherever appropriate, revisions have been made to the manuscript. Responses to these comments and revisions implemented in the paper are detailed below. The Reviewer comments are in italic black fonts while our replies are in blue. The text changed in the manuscript is highlighted in red.
Reviewer # 2
Comments and Suggestions for Authors
(Q1) Grammatical aspects could be improved
(A1) We thank reviewer 2 for this comment. As per the reviewer’s suggestion, the entire manuscript is checked for grammatical/spelling errors, and a number of sentences are revised.
